

# Rate-induced tipping of ice sheets due to visco-elastic Earth response under idealized conditions

Johannes Feldmann[1], Ann Kristin Klose[1,2], Torsten Albrecht[1,3], and Ricarda Winkelmann[1,2,3]

[1]Earth Resilience Science Unit, Potsdam Institute for Climate Impact Research, Member of the Leibniz Association, P.O. Box 60 12 03, 14412 Potsdam, Germany
[2]Institute of Physics and Astronomy, University of Potsdam, 14476 Potsdam, Germany
[3]Department of Evolutionary Earth Systems Science, Max Planck Institute of Geoanthropology, 07745 Jena, Germany

**Correspondence:** Johannes Feldmann (johannes.feldmann@pik-potsdam.de)

**Abstract.** The future evolution of the West Antarctic Ice Sheet may be characterized by self-reinforcing and irreversible retreat due to the unfolding of a marine ice sheet instability (MISI). How the stabilizing mechanism of glacial isostatic adjustment (GIA) does influence the timing and spatial extent of West Antarctica's present and potential future destabilization is highly uncertain and thus increasingly subject of numerical modeling studies that are based on observational data, striving for a most

5 realistic representation of the Antarctic Ice Sheet. Here we employ an ensemble of idealized simulations in a synthetic model setup to systematically investigate how the interaction between ice-sheet dynamics and the visco-elastic response of the solid Earth affect the tipping dynamics of an inherently buttressed, Antarctic-type ice-sheet-shelf system that is perturbed by basal ice-shelf melting. Exploring a wide range of solid Earth structures we find that the threshold of bifurcation-induced tipping (B-tipping), i.e., the critical meltrate magnitude inferred for the ice sheet in quasi equilibrium, strongly depends on the timescale

and the spatial extent of the solid-Earth response. Compared to the case of a fixed bed (no bed deformation), the B-tipping threshold increases for the strongest (East-Antarctic type) Earth structures by at least 80 % (from 0.8 to 1.5 m yr$^{-1}$) whereas for the weakest (West-Antarctic type) Earth structures the increase is more than one order of magnitude larger.

Due to the different timescales involved in the interplay between the dynamics of the ice sheet and the solid Earth, we find that for half of the ensemble members rate-induced tipping (R-tipping) occurs. That is, a sufficiently fast ramp-up of the basal

meltrates triggers a MISI even before the critical forcing threshold of B-tipping would be crossed. In fact, due to R-tipping the effective critical tipping threshold reduces by up to 80 % for high upper-mantle viscosities and thin lithospheres. In none of our simulations bed uplift can stop a MISI once it is triggered, due to the very fast timescale of self-reinforcing grounding-line retreat. Furthermore, we highlight the occurrence of grounding-line overshoots and demonstrate cases of self-sustaining oscillations between advanced and collapsed ice-sheet states. Once triggered, these oscillations continue perpetually just due

to the internal, non-linear interaction between ice-flow and solid-Earth dynamics. Our findings highlight that the character of the solid-Earth structure underlying a MISI-prone ice sheet can strongly affect its tipping dynamics mediated by the strength and timescale of the GIA feedback that counteracts MISI. In the context of Antarctic ice-sheet stability under global warming, our results particularly underscore that besides the *magnitude* also the *rate* of future anthropogenic greenhouse gas emissions are likely to play a crucial role.





## 1 Introduction

Over the last three decades the West Antarctic Ice Sheet (WAIS) has been losing mass steadily, dictating Antarctica's contribution to global sea-level rise (Rignot et al., 2019; Otosaka et al., 2023). This imbalance is mainly driven by increased basal melting of the ice shelves in West Antarctica's Amundsen Sea sector due to enhanced access of relatively warm Circumpolar Deep Water into the ice-shelf cavities (Jenkins et al., 2018; Rignot et al., 2024). The reduced buttressing effect of the thinning ice shelves led to the speed-up, thinning and retreat of the grounded portions of the regional outlet glaciers (Konrad et al., 2018; Milillo et al., 2022), with Pine Island and Thwaites glaciers contributing by far most to the increase in ice discharge into the ocean (Davison et al., 2023b). Numerical modeling suggests that these observed rapid changes might continue into the future (Reese et al., 2023; Bett et al., 2024), possibly leading to a large-scale disintegration of the WAIS (Favier et al., 2014; Joughin et al., 2014; Feldmann and Levermann, 2015; Sutter et al., 2023; Hill et al., 2024; van den Akker et al., 2025) with the potential to increase global mean sea level by $\sim 3$ m (Bamber et al., 2009) and affect the global climate system substantially (Swart et al., 2023; Li et al., 2024) on the long term.

The mechanism underlying a potential WAIS collapse is the so-called Marine Ice Sheet Instability (MISI, Weertman, 1974; Schoof, 2007b): an ice sheet that is grounded on bed below sea level, which is sloping down in landward direction (retrograde bed slope), is prone to self-reinforcing grounding-line (GL) retreat on the retrograde slope due to the highly nonlinear increase of the ice discharge across the GL with increasing bed depth. An increase of the GL flux thins the upstream grounded portion of the ice sheet, which, in turn, induces further GL retreat, closing the self-accelerating MISI feedback loop (Fig. 1). One important stabilizing factor in the MISI context is the buttressing effect of ice shelves that are laterally confined or grounded on topographic highs (Dupont and Alley, 2005; Matsuoka et al., 2015; Fürst et al., 2016; Reese et al., 2018). Their backforce reduces the GL flux, thus alleviating self-sustained GL retreat with the potential to cause GL stabilization on retrograde slope if buttressing is sufficiently strong (Gudmundsson et al., 2012; Cornford et al., 2020; Feldmann et al., 2024). Observations show that West Antarctica's ice shelves generally have been loosing buttressing strength in the recent past and will likely continue to do so in the near future (Gudmundsson et al., 2019; Wild et al., 2022; Miles and Bingham, 2024).

Besides ice-shelf buttressing, the process of glacial isostatic adjustment (GIA) is another major stabilizing factor with potentially strong impact on the occurrence, timing and extent of MISI-type retreat (Gomez et al., 2012; Barletta et al., 2018; Coulon et al., 2021; Book et al., 2022; Albrecht et al., 2024): ice-sheet retreat reduces the ice load on the solid Earth which lifts up in response, decreasing the ocean depth (relative sea level) and thus reducing GL flux which hampers GL retreat (Fig. 1). The strength of this stabilizing GIA feedback depends on the structure of the solid Earth, which determines the rate and length scale of the bed uplift (Adhikari et al., 2014; Larour et al., 2019; Kachuck et al., 2020; Han et al., 2025). Observations suggest that West Antarctica's solid Earth structure is comparatively weak, i.e., it is characterized by an upper-mantle of low viscosity with a thin lithosphere on top, leading to a fast and localized solid-Earth response to ice-load changes, respectively. In contrast, the Earth structure beneath the East Antarctic Ice Sheet is generally found to be much stronger (relatively high upper-mantle viscosity and lithosphere thickness), resulting in a slower and less localized response, with the possibility of strong regional



variations (e.g., Ritzwoller et al., 2001; Lloyd et al., 2020; Wiens et al., 2023; Hansen and Emry, 2025). Note that throughout this study we consider the locality of the bed response as being characterized by the wavelength and the amplitude of the visco-elastic bed deformation. That is, a more localized bed response goes along with a smaller horizontal extent and a larger vertical magnitude of the bed deformation, respectively.

The triggering of a MISI is typically associated with bifurcation-induced tipping (B-tipping), meaning that an ice sheet in equilibrium transitions abruptly and irreversibly from an advanced stable state to a retreated (collapsed) stable state once the control parameter (e.g., basal ice-shelf melting) crosses a critical threshold (Schoof, 2007a; Rosier et al., 2021). The existence of two timescales of (1) ice-sheet retreat and (2) bed response potentially gives rise to a possible *rate dependency* of MISI initiation, i.e., the occurrence of rate-induced tipping (R-tipping; Ashwin et al., 2012; Wieczorek et al., 2023). While the concept of R-tipping has been examined for other tipping elements, such as the Atlantic Meridional Overturning Circulation, or in ecological models (Siteur et al., 2016; Lohmann and Ditlevsen, 2021; Ritchie et al., 2023; Feudel, 2023; Klose et al., 2023; Chapman et al., 2024), to date it remains largely unstudied in the MISI context. Noise-induced tipping is a third possible type of tipping, induced through the (internal) variability of a system (Ditlevsen and Johnsen, 2010; Ashwin et al., 2012).

Here we follow a simplified, yet physically reasonable approach to investigate the first-order interaction between ice and solid-Earth dynamics for a buttressed, Antarctic-type, MISI-prone outlet glacier in an idealized model setup. Sampling a wide range of solid-Earth structures (from very weak to very strong) we systematically examine how bed deformation affects the tipping dynamics of such an idealized ice sheet. This involves inferring the critical B-tipping thresholds of irreversible, MISI-type retreat and re-advance (hysteresis and associated bistability ranges). Another major focus of this study is the investigation of the potential occurrence of R-tipping.

## 2 Methods

### 2.1 Numerical model

To simulate the coupled evolution of the ice-sheet-shelf system and the solid Earth we use the open-source Parallel Ice Sheet Model (PISM; Bueler and Brown, 2009; Winkelmann et al., 2011). Ice flow is computed using a superposition of the shallow-ice approximation (SIA; Morland, 1987) and the shallow-shelf approximation (SSA; Hutter, 1983) of the Stokes stress balance (Greve and Blatter, 2009). In particular, the SSA allows for stress transmission across the GL and therefore accounts for the buttressing effect of ice shelves that are laterally confined (Gudmundsson et al., 2012; Fürst et al., 2016; Reese et al., 2018) and/or in contact with a pinning point (Favier et al., 2012; Matsuoka et al., 2015; Wild et al., 2022; Feldmann et al., 2024). The model applies a linear interpolation of the freely evolving GL and accordingly interpolated basal friction (Feldmann et al., 2014). GL migration has been evaluated in the model intercomparison exercises MISMIP3d (Pattyn et al., 2013; Feldmann et al., 2014) and MISMIP+ (Asay-Davis et al., 2016; Cornford et al., 2020). To improve the approximation of driving stress across the GL, the surface gradient is calculated using centered differences of the ice thickness across the GL (Reese et al.,



2020). All simulations described below are carried out on a regular horizontal grid of 2 km resolution, which has been shown
to be suitable for accurately modeling dynamics of fast-flowing ice in previous studies (e.g., Feldmann and Levermann, 2023).

The response of the solid Earth to ice-load changes is calculated via PISM's built-in "Lingle-Clark" (LC) bed deformation
model (Lingle and Clark, 1985; Bueler et al., 2007). The two-layer LC model assumes a Maxwell rheology and is characterized
by an Elastic plate Lithosphere that is situated above a half-space representing the Viscous Astenosphere (thus termed ELVA in
Swierczek-Jereczek et al., 2024). The resulting visco-elastic response of the solid Earth to ice-load changes hence consists of
an instantaneous and a time-dependent part, the latter one being associated with the prescribed viscosity $\eta$ of the upper mantle
(see Appendix A). The thickness of the elastic lithosphere $T_e$ is the second central parameter of the ELVA model, affecting the
spatial distribution of the response. Both parameters neither vary laterally nor vertically, i.e., they are spatially uniform, and
they are also held fixed in time. The LC module is updated at an interval of 10 model years. No sea-level model is applied, i.e.,
the height of the sea surface is kept constant in all simulations. Furthermore, rotational effects are neglected in the model.

## 2.2 Setup and experimental design

### 2.2.1 Initial bed topography and boundary conditions

Our model setup is designed to simulate the self-reinforcing retreat of an idealized, inherently buttressed Antarctic-type outlet
glacier in interaction with the visco-elastic response of the underlying solid Earth. For this purpose we initialize the model
with a modified version of the channel-type MISMIP+ bed topography, as used in Feldmann and Levermann (2023, Appendix
C), which is held fixed during the model spinup (Fig. 2a). The qualitative shape of the initial bed topography inside the bed
trough is as follows (Fig. 2b): along the setup centerline, the bed elevation drops from an inland sill (located at $x = 0$) into an
overdeepening (deepest point of the bed depression at $x_{BD} = 200$ km) and increases towards a coastal sill further downstream
(highest point at $x_{CS} = 500$ km) beyond which the bed elevation drops again (continental shelf break). The initial depth of
the overdeepening between $x_{BD}$ and $x_{CS}$ is $D_{BD} = 900$ m. The resulting average retrograde bed slope of $3 \cdot 10^{-3}$ is on the
same order of magnitude as the observed slope steepness of Thwaites Glacier over the same length scale. The bed trough that
confines the fast-flowing ice stream (Figs. 2a and S1) has a width of 160 km, making it similarly wide as Thwaites Glacier's
observed fast-flowing trunk at the GL (e.g., Davison et al., 2023a, Fig. 1a).

The surface accumulation rate is prescribed as a constant in space and time (see Table 1, also for other parameters). The
same applies to the ice softness, i.e., the ice is assumed to be isothermal, neglecting thermomechanical coupling. The basal
ice-shelf meltrate is spatially uniform, being zero during model spinup while varying in time in subsequent experiments. Basal
melting of grounded ice is neglected. Prescribing a spatially and temporally uniform friction coefficient in a Weertman-type
friction law (Cornford et al., 2020, Eq. 3 of), the resulting basal stresses vary over space and time as part of the non-local SSA
stress balance (see Discussion). A single calving condition is applied, where all ice moving beyond $x_{CF} = 780$ km is calved
off. Regarding the lateral margins of the computational domain, i.e., $y = \pm 160$ km, periodic boundary conditions apply. Note
that we avoid prescribing a boundary condition at $x = 0$ by mirroring the setup (including the sub-ice-shelf melt perturbations)





along this axis, producing an ice sheet that consists of two symmetric outlet glaciers that share the same ice divide (as we have done in previous studies, e.g., Cornford et al., 2020; Feldmann et al., 2022, 2024). The entire computational domain thus spans from $-800$ to $+800$ km in the $x$ direction. Due to the symmetry of the system with respect to the ice divide, we here only consider one of the two outlet glaciers, i.e., the right-hand half of the computational domain ($x \geq 0$).

### 2.2.2 Model spinup

The spinup starts from a block of ice of 2000 m thickness on the prescribed bed topography. From this initial state an ice-sheet-shelf system evolves that reaches equilibrium after several 10,000 model years. The resulting ice sheet is drained by an ice stream (Fig. S1) through the bed trough, feeding a bay-shaped ice shelf which provides buttressing to the upstream ice. The depth of the initial bed geometry was chosen such that the steady-state GL of the ice stream inside the bed trough is located on the tip of the coastal sill at the end of the spinup (Fig. 2). This way, in the course of the subsequent perturbation experiments, a relatively small increase in basal ice-shelf melting $m$ (from 0 to $\approx 1 \, \mathrm{m \, yr^{-1}}$) can push the GL from its stable position towards an unstable position on the retrograde slope, triggering a MISI-type retreat of the ice sheet. Since the deformation of the bed is switched off during the spinup, the resulting steady-state ice sheet can be considered to be in equilibrium with the prescribed bed topography. In fact, enabling bed deformation after the system has equilibrated does not lead to any changes in the ice and bed geometries.

### 2.2.3 Ensemble of solid Earth structures

The equilibrium obtained from the spinup serves as the initial state for an ensemble of simulations that systematically explores how the configuration of the solid-Earth structure affects the interaction between the dynamics of the ice sheet and the solid Earth. For this purpose, we vary the two central parameters that determine the solid-Earth structure in the LC model, i.e., the upper-mantle viscosity, $\eta$, and the lithosphere thickness, $T_e$ (see Table 1). In each simulation both parameters are constant in space and time. Our selection of the parameter values ($\eta = \{10^{18}, \, 5 \cdot 10^{19}, \, 10^{21}, \, 5 \cdot 10^{22}\} \, \mathrm{Pa \, s}$ and $T_e = \{20, \, 80, \, 140, \, 200\}$ km) is based on the parameter ranges used by Coulon et al. (2021), liberally covering the observed ranges and the associated uncertainties of these two variables (e.g., Ritzwoller et al., 2001; Whitehouse et al., 2019; Lloyd et al., 2020; Wiens et al., 2023). The parameter space resulting from each combination of the above values of $\eta$ and $T_e$ (16 combinations in total) in particular captures the weaker Earth structures (low $\eta$ and $T_e$ values) that are characteristic for West Antarctica, and the stronger Earth structures (high $\eta$ and $T_e$ values) that are characteristic for East Antarctica. Furthermore, the parameter space represents intermediate solid Earth structures (medium values of $\eta$ and $T_e$) as well as less typical combinations (high values of $\eta$ combined with low values of $T_e$ and vice versa).

### 2.2.4 Ramp-up/down (hysteresis) experiments

All 16 members of the solid-Earth parameter space (plus one model realization with a fixed bed) start from the same ice-sheet-shelf system that has been spun up on the same fixed bed topography. In each of these realizations the initial equilibrium is





perturbed by basal ice-shelf melting, that, starting from zero, is ramped up at a rate of $\dot{m} = 10^{-5} \ \mathrm{m\,yr}^{-2}$ (red line in Fig. 3). The previously fixed bed topography is now free to evolve. The very slow increase in sub-ice-shelf melting ($1 \ \mathrm{m\,yr}^{-1}$ every 100,000 years) is intended to let the system evolve in a quasi equilibrium. Note that the forcing rate is reduced by one order of magnitude for all runs with the largest upper-mantle viscosity ($\eta = 5 \cdot 10^{22} \ \mathrm{Pa\,s}$) due to the associated very slow response time of the solid Earth (Table 2). In most cases, the meltrate ramp-up ends at a final value of $5 \ \mathrm{m\,yr}^{-1}$. By then, the ice sheet has collapsed and stabilized. Subsequently, the forcing is reversed, i.e., the meltrate is ramped down at the same slow rate (blue line in Fig. 3). During this ramp-down the ice sheet re-advances and tips back to its original advanced state. Note that for three of the weakest Earth structures ($\eta = 10^{18}$, $5 \cdot 10^{19}$ or $10^{21} \ \mathrm{Pa\,s}$ with $T_e = 20 \ \mathrm{km}$) a higher final forcing magnitudes (ramp-up to $15 \ \mathrm{m\,yr}^{-1}$ instead of $5 \ \mathrm{m\,yr}^{-1}$, but at the same rate) had to be applied in order to capture ice-sheet collapse and subsequent stabilization. In general, the ice-sheet response (e.g. GL migration) to the ramp-up and subsequent ramp-down forms a hysteresis curve (Fig. 4a). In the following, we refer to the GL curve in response to the slow ramp-up (ramp-down) as the forward (backward) branch of the hysteresis curve. Most of the analysis in this study will focus on the forward hysteresis branch, motivated by the observed recent and present-day retreat and potential future collapse of the MISI-prone WAIS.

### 2.2.5 Branch-off experiments to infer B-tipping thresholds

To infer the critical forcing magnitude at which ice-sheet collapse (or re-advance) occurs, we run simulations that branch off from the hysteresis curve at different meltrates prior to collapse (or re-advance), letting the system run into equilibrium under the respective constant melt rate (red and blue lines with triangle in Fig. 3). For the forward branch of the hysteresis curve (GL retreat), the critical threshold of B-tipping is located between the largest forcing magnitude for which the glacier still stabilizes in its advanced state and the next larger forcing magnitude for which the glacier collapses. The branch-off experiments are also carried out for the backward branch of the hysteresis curve. Here, the critical threshold of B-tipping lies between the smallest meltrate for which the glacier still stabilizes in its collapsed state and the next lower meltrate for which the glacier tips back into the advanced stable regime. The branch-off simulations are sampled at a meltrate interval of $0.1 \ \mathrm{m\,yr}^{-1}$. For simplicity, we will refer to the critical meltrate at which tipping occurs, $m_{\mathrm{B-tip}}$, as the lowest (highest) meltrate for which the ice sheet collapses (re-grows) in the branch-off experiments.

### 2.2.6 Step-forcing experiments to detect potential R-tipping

To investigate the potential occurrence of R-tipping for the forward branch of the hysteresis curve, the initial steady state is perturbed by a step forcing to a meltrate level just below the B-tipping threshold for each model realization. That is, the forcing is ramped up from zero to the specific meltrate instantaneously (red line with circle in Fig. 3). If this results in a collapse of the ice sheet, then the model realization exhibits R-tipping due to the fact that a slow ramp-up to the same meltrate does not lead to a collapse. In this case, further step-forcing experiments to lower meltrate levels are conducted to confine the range of R-tipping occurrence associated with the specific model realization. These experiments are carried out analogously for the backward branch of the hysteresis curve (blue line with circle in Fig. 3). Starting from the initial state of the backward branch, the meltrate is instantaneously lowered from the initial value (typically $5 \ \mathrm{m\,yr}^{-1}$) to meltrates above the backward B-tipping





threshold in order to check whether the ice sheet tips back into its advanced state for this instantaneous ramp-down (potential occurrence of backward R-tipping).

## 3 Results

### 3.1 Hysteresis and involved feedbacks

The slow ramp-up of the forcing causes a gradual retreat of the ice sheet's GL onto the retrograde bed slope: the applied melting
at the ice-shelf base thins the ice shelf, which reduces its buttressing effect, inducing higher ice discharge across the GL and thinning of the upstream grounded ice sheet, resulting in GL retreat (Figs. 1 and 4a,b). Since the retrograde bed slope is very shallow in the vicinity of the tip of the costal sill (initial location of the GL; Fig. 2), the destabilizing MISI mechanism of self-reinforcing GL retreat (Fig. 1) is weak initially. Thus, in the absence of bed deformation, the buttressing effect of the ice shelf is large enough to stabilize the GL on the shallow retrograde slope for low forcing magnitudes. In case of a deformable
bed, the bed uplift in response to the decreasing ice load leads to a decrease in the relative sea level, reducing the ice discharge across the GL (stabilizing GIA feedback in Fig. 1), thus allowing stable GL positions farther inland and for higher forcing magnitudes (Figs. 4 and 5). In most cases, the MISI becomes dominant when the GL approaches $x = 400$ km. Then the strong increase in GL flux (due to the increase in GL bed-depth in the course of GL retreat) cannot be compensated by the uplifting bed such that the MISI fully unfolds.

As the ice sheet undergoes rapid retreat down the retrograde slope, the GL passes through the bed depression and enters the inland prograde bed section, which acts to stabilize the GL due to the decrease in GL flux with GL retreat. However, a direct stabilization of the GL on the prograde slope does only occur in the presence of a sufficiently weak Earth structure and thus a strong GIA feedback (Fig. 5), i.e., for the combination of a sufficiently localized response ($T_e \leq 140$ km) and a sufficiently fast response time ($\eta \leq 5 \cdot 10^{19}$ Pa s). In all other cases, the GL retreats all the way to $x = 0$ such that the ice stream inside the
bed trough vanishes (temporarily). Several model realizations show an overshoot of the GL, i.e., after retreating to $x = 0$ the GL re-advances in response to the delayed bed uplift (see Sect. 3.5). For the most widespread and thus least localized response (thickest lithosphere, $T_e = 200$ km) the GL does not re-advance at all. Irrespective of the occurrence of GL overshoot, thinner lithospheres generally promote GL stabilization further downstream due to the stronger locality of the bed response.

In the course of ice-sheet retreat, collapse and stabilization, the bed elevation overall increases by several $100$ m in the
previously grounded area compared to the initial state (Fig. 4b). The bed uplift is most pronounced in the region of the bed depression, i.e., the region of the initially thickest ice (local uplift of up to nearly $1,000$ m; Fig. S2a). As a consequence, the retrograde bed slope flattens (Fig. 4b). The ramp-down of the forcing then causes a slow regrowth of the ice sheet as the ice shelf thickens and thus provides more buttressing. The advance of the ice sheet goes along with bed subsidence due to the increasing ice load (Fig. 4c), hampering GL advance. However, at some stage the basal-melt forcing has decreased sufficiently
such that the thickened ice shelf provides enough buttressing to induce GL advance onto the retrograde bed section. At this





point GL advance becomes self-reinforcing due to the strong inverse MISI feedback, which outpaces the stabilizing GIA feedback. Eventually, the GL reaches its initial position when the meltrate has been ramped down to zero (initial value). Note that for a solid-Earth response that spreads comparatively wide ($T_e \geq 140$ km), GL re-advance towards its initial position can occur in several steps (blue curves in Fig. 5a-h) as opposed to the usual large, single-step event of GL migration after passing the tipping threshold. This is a result of the temporary grounding (pinning) of the thickening ice shelf on the retrograde slope in the course of slowly increasing meltrates. Such temporary grounding has been reported in earlier idealized studies for both MISI-type ice-sheet retreat (Jong et al., 2018) and re-advance (Feldmann et al., 2024). In our simulations the bed subsidence emerging from the ice-shelf grounding hampers GL re-advance such that the full re-advance occurs for comparatively low (even negative) forcing values.

## 3.2 B-tipping

The meltrate threshold, $m_{\mathrm{B-tip}}$, at which the destabilization (MISI) is triggered during the forcing ramp-up strongly depends on the configuration of the solid-Earth structure. The lowest B-tipping value, $m_{\mathrm{B-tip,fixed}} = 0.8$ m yr$^{-1}$, is found for the fixed-bed case due to the absence of bed uplift that could counteract GL retreat (Fig. 5a, dashed grey line). If the bed is allowed to adjust to ice-load changes, then weaker Earth structures (lower values of the mantle viscosity and/or the lithosphere thickness) generally shift the B-tipping threshold towards higher values (going from the upper right to lower left in Fig. 5). The associated generally faster and more localized bed-uplift rates have a larger effect in diminishing GL ice discharge and thus GL retreat. To express the change in the B-tipping threshold relative to the reference case of a fixed bed, we calculate

$$\Delta m_{\mathrm{B-tip}} = \frac{m_{\mathrm{B-tip}} - m_{\mathrm{B-tip,fixed}}}{m_{\mathrm{B-tip,fixed}}} \cdot 100 \%, \tag{1}$$

with critical B-tipping thresholds of the specific solid-Earth configuration, $m_{\mathrm{B-tip}}$, and of the fixed-bed case, $m_{\mathrm{B-tip,fixed}}$ (numbers in brackets in Fig. 6a). Thus, a doubling of the B-tipping threshold relative to the fixed-bed case would mean a shift of $\Delta m_{\mathrm{B-tip}} = +100$ %. For a lithosphere thickness of $T_e \geq 80$ km, the critical forcing magnitude is doubled or tripled compared to the fixed-bed case (increase of B-tipping threshold ranges from $\Delta m_{\mathrm{B-tip}} = 87$ to $225$ %; Fig. 6a). For the lowest $T_e$ value, however, the shift in the B-tipping threshold is up to an order of magnitude larger, ranging from $325$ % to $1462$ % (purple regimes in Fig. 6a), which results from the strong locality of the response associated with the very thin lithosphere.

In all configurations of the solid-Earth structure the ice-sheet collapse in response to the forcing ramp-up is irreversible, as the re-advance of the ice sheet during the subsequent forcing ramp-down does not take place on the same path as the retreat (Figs. 4a and 5). In fact, the return of the ice sheet to its advanced state requires a lowering of the forcing magnitude below the threshold of ice-sheet collapse. The resulting hysteresis curve of, e.g., the GL position envelopes a bistability regime in which the ice sheet can be either in its advanced or collapsed state, depending on the forcing history. The size of the bistability range increases for increasing weakness of the solid-Earth structure, resulting from the associated increasing strength of the stabilizing GIA feedback, which shifts the thresholds of (1) collapse to higher values and (2) re-advance to lower values.





### 3.3 R-tipping

For several configurations of the solid-Earth structure the threshold of triggered self-reinforcing retreat depends on the rate of the increase in basal ice-shelf melting (R-tipping; red shading in Figs. 4a and 5). In these cases, a sufficiently fast forcing rate
induces an ice-sheet collapse at a forcing magnitude that is below the critical threshold of B-tipping (see Movie S1). Analogous to the case of B-tipping (Eq. 1), we express the rate-dependent shift in the critical tipping threshold compared to the fixed-bed case by

$$\Delta m_{\mathrm{R-tip}} = \frac{m_{\mathrm{R-tip}} - m_{\mathrm{B-tip,fixed}}}{m_{\mathrm{B-tip,fixed}}} \cdot 100 \ \%, \tag{2}$$

where $m_{\mathrm{R-tip}}$ is the R-tipping threshold for an infinitely fast forcing ramp-up (lower end of the R-tipping regime; see Fig. 4a).
For the occurrence of R-tipping, the shift in the tipping threshold relative to the fixed-bed case is roughly only half as large or even smaller compared to the case of B-tipping. The resulting ratios of $\Delta m_{\mathrm{R-tip}}/\Delta m_{\mathrm{B-tip}}$ range from $15 \ \%$ to $57 \ \%$ (generally decreasing for thinner lithospheres and larger upper-mantle viscosities; Fig. 6b), indicating an efficient reduction of the critical tipping threshold through the occurrence of R-tipping.

Throughout the ensemble, R-tipping occurs for intermediate to high values of $\eta$, i.e., for moderate to high strengths of the
Earth structure. This can be explained by considering (1) the timescale of the viscous solid-Earth response, $\tau_{\mathrm{SE}}$, which increases linearly with $\eta$ (see Appendix A), and (2) the timescale of stable GL retreat (ice-load change) prior to tipping that would apply in the absence of bed deformation, $\tau_{\mathrm{L,st}}$, which decreases for a higher forcing rate (decreasing ice-shelf buttressing) and for a steeper retrograde bed slope (larger GL flux). In the following we distinguish between three cases to argue under which conditions R-tipping generally occurs (visualized in Fig. 7):

1. **Very low mantle viscosity** (no R-tipping, Fig. 7a): the solid-Earth response is very fast compared to the timescale of stable GL retreat (ice-load change), i.e., $\tau_{\mathrm{SE}} << \tau_{\mathrm{L,st}}$ (in the limit case the response would be instantaneous). Thus, the bed adjusts quasi immediately to GL retreat, regardless of the timescale of the forcing. This means that the bed is close to equilibrium with respect to the changing ice load even for a high forcing rate and thus a comparatively fast, stable GL retreat prior to the ice-sheet collapse. Consequently, the threshold of destabilization is independent of the forcing rate.

2. **Moderate to high mantle viscosity** (R-tipping occurs, Fig. 7b): the timescale of the solid-Earth response is similar to or slower than the timescale of GL retreat. Under this condition it is possible that the bed is close to equilibrium with respect to the changing ice load for a very low forcing rate ($\tau_{\mathrm{SE}} < \tau_{\mathrm{L,st}}$) but not for a high forcing rate ($\tau_{\mathrm{SE}} > \tau_{\mathrm{L,st}}$). The lagged solid-Earth response to a high forcing rate (and thus faster retreat) leads to ice-sheet destabilization at lower forcing magnitudes (R-tipping threshold) compared to the B-tipping threshold, which holds for a very low forcing rate
(and thus slower retreat).

3. **Fixed bed** (no R-tipping, Fig. 7c): the solid-Earth does not respond to the ice-load change, i.e. the timescale of the response can be considered as infinitely slow ($\tau_{\mathrm{SE}} \to \infty$). Thus, regardless of the forcing rate, the GL retreat and bed evolution do not interfere, making the threshold of destabilization independent of the forcing rate.



Note that the locality of the solid-Earth response modulates the pattern of R-tipping occurrence described above: for a given upper-mantle viscosity a more localized bed uplift due to a thinner lithosphere is associated with a higher B-tipping threshold, i.e., a higher critical basal meltrate. Larger basal meltrates and associated stronger ice-shelf thinning reduce the ice-shelf buttressing effect and force the GL into regions of deeper and steeper bed slope. Both effects increase GL discharge rates and thus reduce the timescale of GL retreat (smaller $\tau_{\mathrm{L,st}}$). For instance, in the case of $\eta = 5 \cdot 10^{19}\ \mathrm{Pa\,s}$ the timescale of the solid-Earth response, $\tau_{\mathrm{SE}} = 350\ \mathrm{yr}$, is generally faster than the timescale of GL retreat, $\tau_{\mathrm{L}} \sim \tau_{\mathrm{L,st}} \approx 6000\ \mathrm{yr}$, preventing the occurrence of R-tipping (see Table 2 for more numbers). However, for the thinnest lithosphere, the strong forcing magnitude required to trigger the MISI decreases $\tau_{\mathrm{L,st}}$ substantially, leading to a delay in the solid-Earth response ($\tau_{\mathrm{L,st}} < \tau_{\mathrm{SE}}$) that allows for R-tipping (Fig. 5n). This also explains the presence (absence) of R-tipping for a thin (thick) lithosphere in case of $\eta = 10^{21}\ \mathrm{Pa\,s}$. For this viscosity value the timescales of stable GL retreat ($\tau_{\mathrm{L,st}} \approx 6{,}000\ \mathrm{yr}$) and solid-Earth response ($\tau_{\mathrm{SE}} \approx 7{,}000\ \mathrm{yr}$) are generally roughly similar. However, the smaller tipping thresholds for thicker lithospheres result in timescales of stable GL retreat that are effectively slower (larger $\tau_{\mathrm{L,st}}$), allowing for a sufficiently fast adjustment of the solid Earth to ice-load changes and thus preventing R-tipping (Figs. 5c and g).

Note that for some model realizations R-tipping is also present for the backward branch of the hysteresis. However, in no occasion the R-tipping range is larger than $0.1\ \mathrm{m\,yr^{-1}}$, i.e., the sampling rate of our analysis, which is why we treat these occurrences as negligible.

### 3.4 Influence of bed-depression depth

Besides the solid-Earth structure, the thresholds of B-tipping and R-tipping are also affected by the initial depth of the bed depression (Fig. 2b): on a shallower retrograde bed slope each increment of GL retreat causes a smaller increase in GL bed depth (and thus in GL ice discharge) such that the system enters the unstable regime at higher absolute forcing magnitudes compared to the case of a steeper retrograde slope. However, for the major part of the parameter space ($T_e \geq 80\ \mathrm{km}$) the shift in the B-tipping threshold with respect to the fixed-bed case is roughly the same when comparing pairs of ensemble members that only differ in the initial slope steepness (maximum deviation of $\Delta m_{\mathrm{B-tip}}$ value is $10\ \%$; compare Figs. S4a and b). Larger differences occur for the thinnest lithosphere for which the increase in the B-tipping threshold relative to a fixed bed ($\Delta m_{\mathrm{B-tip}}$ value) is about $55\ \%$ to $126\ \%$ stronger in the shallow-slope case compared to the steep-slope case. Thus, the shift in the critical B-tipping threshold is qualitatively the same for both initial bed geometries and relevant quantitative differences only occur for the thinnest prescribed lithosphere. A similar pattern applies to the case of R-tipping (compare Figs. S4c and d).

### 3.5 GL overshoots

Most of the model realizations exhibiting R-tipping show a GL overshoot following the triggered MISI. That is, in these runs the GL retreats back to $x = 0$ before it re-advances by several $10\ \mathrm{km}$ (up to $150\ \mathrm{km}$) on the prograde slope of the central sill where the GL stabilizes eventually (Fig. 5). Such GL overshoots in response to lagged bed rebound have been reported by Kingslake et al. (2018) in paleo reconstructions of the WAIS evolution during the Holocene. In our simulations the re-advance





is initiated once the bed depth in the vicinity of $x = 0$ is shallow enough to allow for the advance of grounded ice from the lateral ridges towards the center of the bed trough (Fig. 4b). The duration (timescale) of the overshoot strongly depends on the upper-mantle viscosity: for $\eta = 10^{21}$ Pa s the return of the GL takes about $10,000$ yr whereas for $\eta = 5 \cdot 10^{22}$ Pa s it takes at least $300,000$ yr – in line with the timescaling expected from theory (see Appendix A and Table 2). In most cases of GL overshoot occurrence during the forcing ramp-up, it also occurs for the ramp-down (though at a smaller magnitude). That is, after passing the retrograde slope in downstream direction, the GL first advances beyond the tip of the sill before retreating back onto the tip of the sill, reaching its original (initial) equilibrium position. In the following we describe the conditions which favor a GL overshoot in the course of ice-sheet collapse in our simulations:

1. **Overshoot predominantly occurs for medium to high mantle viscosity** ($\eta \geq 10^{21}$ Pa s). This is due to the fact that an overshoot emerges from the delay of the full response of the solid Earth to the ice-load reduction in the course of GL retreat. Thus, the bed uplift rate is too small to directly stabilize the relatively quickly retreating GL on the prograde slope in the aftermath of the MISI. Consequently, the GL reaches $x = 0$ temporarily and GL re-advance sets in once the delayed bed uplift allows for the re-advance of grounded ice from the lateral margins into the central bed trough. For lower mantle viscosities ($\eta \leq 5 \cdot 10^{19}$ Pa s) the relatively fast solid-Earth response prevents such delay and thus no overshoot occurs. An exception to this is given by the value pair $\eta = 5 \cdot 10^{19}$ Pa s, $T_e = 20$ km, in which case the very localized bed response requires a relatively strong forcing magnitude to trigger the MISI, leading to faster GL retreat during collapse, and thus allowing for some delay in the solid-Earth response (see Sect. 3.3) which results in a rather small overshoot (GL not reaching $x = 0$).

2. **Overshoot is more pronounced for thinner lithospheres**. The more localized response of the solid Earth associated with smaller $T_e$ leads to a stronger overall bed uplift in the vicinity of the GL and thus promotes post-collapse GL stabilization in a more advanced position. This causality applies generally for all conducted simulations, independently of whether overshoot occurs or not (increase in final GL position after collapse when going from top to bottom in Figs. 5 and S3) and in particular determines the size (spatial dimension) of a potential overshoot for a given mantle viscosity.

3. **Overshoot is stronger for a shallower bed depression**, i.e., a flatter prograde slope of the central sill. While a flatter slope eases the complete retreat of the GL to $x = 0$ during collapse it also promotes GL re-advance afterwards, leading to an eventually more advanced GL position (Fig. S3).

The reason why R-tipping and GL overshoot occur together in most cases lies in the fact that both phenomena require a delay in the viscous solid-Earth response with respect to ice-load changes, i.e., $\tau_{\mathrm{SE}} < \tau_{\mathrm{L}}$. In line with the three conditions listed above, in our simulations the strongest GL overshoot ($\approx 150$ km) occurs for a medium to high mantle viscosity ($\eta \geq 10^{21}$ Pa s), a very thin lithosphere ($T_e = 20$ km) and a shallow bed depression ($D_{\mathrm{BD}} = 300$ m).



## 3.6 Self-sustaining oscillations

The two model configurations associated with the largest GL overshoots mentioned above allow for the emergence of self-sustaining oscillations between ice-sheet collapse and regrowth (Fig. S3o,p): in this case the GL overshoot after the collapse is so strong that it induces ice-sheet re-advance past the bed depression, leading to a temporary, almost complete re-advance of the ice sheet. The induced bed subsidence, in turn, triggers the next phase of ice-sheet collapse, completing the cycle (Fig. 9, Movie S2). Consequently, once triggered by crossing a critical meltrate threshold (forcing can be held constant afterwards), alternating ice-sheet collapse and re-growth continue perpetually just due to the interaction between the dynamics of the ice sheet and the solid Earth. The full forward and backward branches of GL migration are shown in Fig. S3o (model configuration: $\eta = 10^{21}$ Pa s, $T_e = 20$ km, $D_{\mathrm{BD}} = 300$ m): under the continuation of the slow forcing ramp-up, the self-sustaining oscillations persist from a forcing magnitude of $13.5$ m yr$^{-1}$ up to $23.5$ m yr$^{-1}$, beyond which the system eventually stabilizes in the collapsed state. For the same model configuration the slow forcing ramp-down yields similar self-sustaining oscillations regarding amplitude, wavelength and meltrate range. Note that for the second model configuration exhibiting self-sustained oscillations (Fig. S3p) the hysteresis branch is incomplete due to the extremely low rates of forcing ramp-up applied in this case of very high upper-mantle viscosity ($\eta = 5 \cdot 10^{22}$ Pa s) and the associated very long computational time (factor ten compared to the case of $\eta = 10^{21}$ Pa s).

## 4 Discussion

Conducting idealized ensemble simulations, we systematically explored the tipping dynamics of a simplified, Antarctic-type, inherently ice-shelf buttressed, MISI-prone outlet glacier (Fig. 2) in interaction with the visco-elastic deformation of its underlying bed for a wide range of solid-Earth structures (here as combinations of upper-mantle viscosity $\eta$ and elastic lithosphere thickness $T_e$; see Table 1). According to observations, larger (smaller) upper-mantle viscosities are typically associated with larger (smaller) lithosphere thicknesses (e.g., Coulon et al., 2021) and thus parameter combinations for $\eta$ and $T_e$ that are situated along the diagonal of our parameter space can be considered to be most realistic (Figs. 5 and S3). Consequently, solid-Earth characteristics that are broadly representative of West Antarctic (WA) conditions are found in the lower-left of the parameter space, while the ones representing East Antarctic (EA) conditions are found in the upper-right.

According to our results, the critical B-tipping thresholds of MISI-type ice-sheet collapse increase with increasing weakness of the solid-Earth structure, i.e., the external basal ice-shelf melt forcing has to be ramped up to higher magnitudes to trigger the self-reinforcing retreat. This means that the WA solid-Earth parameter subspace shows a much stronger potential to counteract self-accelerating ice-sheet retreat than the EA subspace. These findings are consistent with outcomes from various previous studies that highlighted the potential of weaker Earth structures to delay (West) Antarctic ice-sheet retreat (e.g. Gomez et al., 2012; Larour et al., 2019; Kachuck et al., 2020; Coulon et al., 2021; Gomez et al., 2024; Han et al., 2025). The idealized approach taken in this study allowed us to prepare a model setup in which the initial steady state of the outlet glacier is close to the glacier's B-tipping point in the reference case of a fixed bed. This means that a slow ramp up of ice-shelf basal



melting from 0 to $0.8 \, \mathrm{m \, yr^{-1}}$ is sufficient to trigger MISI-type ice-sheet collapse. For the stronger (EA) solid-Earth structures in our ensemble, this B-tipping threshold is roughly doubled or tripled (Eq. 1; Figs. 6 and S4). For the weakest (WA) solid-

375 Earth structures, however, the threshold increase is one order of magnitude larger. The strong increase of $\Delta m_{\mathrm{B-tip}}$ towards weaker Earth structures in the parameter space results from the combination of (1) the faster timescale of bed deformation for smaller upper-mantle viscosities (see Appendix A and Table 2) and (2) the more localized bed uplift for thinner lithospheres. The resulting comparatively large bed uplift rates at the GL allow for a strong GIA feedback and thus a very effective MISI suppression (Figs. 1 and S2a). In contrast, towards stronger solid-Earth structures, the bed uplift is more widespread and thus

weaker at the GL (Fig. S2b-d), making MISI suppression less efficient.

Our simulations furthermore demonstrate the occurrence of R-tipping in the MISI context (Fig. 4), resulting from a separation of timescales in the interaction between ice sheet and solid Earth (Fig. 7): if the solid Earth cannot adjust quickly enough to ice-load changes then a sufficiently fast rate of the external forcing (basal ice-shelf melting) ramp-up leads to a MISI trigger at a forcing threshold lower than the B-tipping threshold (Movie S1). The relatively quick response times of the weak solid-Earth

structures that represent WA bed characteristics prevent R-tipping in most instances of this parameter subspace (Figs. 6b and 7a). Conversely, the (very) slow response timescales that are more characteristic for the comparatively strong EA solid-Earth structures allow for the occurrence of R-tipping (Fig. 7b). This is in accordance with results from real-world simulations from Swierczek-Jereczek et al. (2025) who find that the EAIS is much more likely to undergo R-tipping than the WAIS for forcing (atmospheric and oceanic warming) rates that are smaller than observed present-day values. The same holds true for the applied

ramp-up rates of basal ice-shelf melting that lead to R-tipping in our synthetic setup. Depending on the upper-mantle viscosity, the slowest ramp-up rates required to induce R-tipping range from about $\dot{m}_{\mathrm{R-tip}} \sim 10^{-4} \, \mathrm{m \, yr^{-2}}$ (for $\eta = 10^{21} \, \mathrm{Pa \, s}$; Figs. 8 and S7) to $\dot{m}_{\mathrm{R-tip}} \sim 10^{-2} \, \mathrm{m \, yr^{-2}}$ (for $\eta = 5 \cdot 10^{19} \, \mathrm{Pa \, s}$; Figs. S5 and S6), i.e., the uniform sub-ice-shelf meltrates increase by $\sim 0.001$ to $0.1 \, \mathrm{m \, yr^{-1}}$ per decade. Note that the scaling of $\dot{m}_{\mathrm{R-tip}}$ with the prescribed viscosity values is in line with the theoretical linear relation between viscosity and associated Maxwell time (Appendix A). The observed long-term temperature

trend of Circumpolar Deep Water masses entering the continental shelf of West Antarctica's Amundsen Sea Sector is about $0.2 \, \mathrm{K \, yr^{-1}}$ per decade (Schmidtko et al., 2014). This may result in an increase in the average sub-ice-shelf meltrates on the order of $\sim 1 \, \mathrm{m \, yr^{-1}}$ per decade (according to an observed simple linear relation from Rignot and Jacobs, 2002, see Fig. 2 therein). That means that the observed rates of meltrate increase in West Antarctica are generally by one or several orders of magnitude higher than the minimal rates required to effectuate R-tipping in our simulations. This suggests that analyses of

the stability of the Antarctic Ice Sheet should take into account the possibility of R-tipping, e.g., in projections of the future ice-sheet evolution under different forcing scenarios. Note that the above estimates of ice-shelf meltrate change in the Amundsen Sea are rather conservative as meltrates might increase faster due to possible feedbacks and nonlinearities in the ice-ocean interaction (e.g. Holland et al., 2023; Kreuzer et al., 2025).

In the event of R-tipping, the increase in the tipping threshold relative to the fixed-bed case, $\Delta m_{\mathrm{R-tip}}$ (Eq. 2), is typically

at least $20 \, \%$ smaller than in the case of B-tipping (Fig. S4). This reduction of the effective tipping threshold for larger forcing rates generally strengthens with increasing $\eta$, which is in accordance with findings from Swierczek-Jereczek et al. (2025) which




are based on numerical experiments that vary Antarctic upper-mantle viscosities but leave the field of the lithosphere thickness fixed in all simulations. Altering the lithosphere thickness in our simulations, we find a reduction of the effective tipping threshold for decreasing $T_e$, i.e., a stronger increase of the B-tipping threshold $\Delta m_{\mathrm{B-tip}}$ compared to the R-tipping threshold

$\Delta m_{\mathrm{R-tip}}$, leading to larger R-tipping ranges in Figs. 5 and S3. In this regime of stronger basal melting the destabilizing MISI feedback is enhanced through the strong reduction of ice-shelf buttressing and GL retreat into regions of steeper retrograde bed slope, resulting in potentially faster GL retreat timescales that lead to tipping at lower basal meltrates if the rate of meltrate increase is sufficiently fast. Our ensemble-wide minimum value of the ratio $\Delta m_{\mathrm{R-tip}}/\Delta m_{\mathrm{B-tip}}$ is $\approx 0.05$ (for $\eta = 5\cdot10^{22}\,\mathrm{Pa\,s}$, $T_e = 20$ km and $D_{\mathrm{BD}} = 300$ m), meaning that the forcing magnitude required to induce R-tipping is only a twentieth of the

corresponding forcing magnitude required to induce B-tipping. However, this parameter combination, located in the lower-right corner of the parameter space, represents rather unrealistic solid-Earth conditions. More realistic parameter combinations (closer to the diagonal of the parameter space) yield a range of ratios from $\Delta m_{\mathrm{R-tip}}/\Delta m_{\mathrm{B-tip}} \approx 0.4$ to $0.8$ corresponding to a tipping threshold reduction of 20 to 60 %.

Occurrences of R-tipping that we detected for the backward branch of the hysteresis are discarded since in no case the R-

420 tipping range is larger than the sampling rate of $0.1\,\mathrm{m\,yr^{-1}}$ used to infer B-tipping thresholds. Moreover, comparability within our ensemble would be limited due to the fact that the backward branches of the different model realizations start from different ice-bed geometries and different meltrates (ranging from $3\,\mathrm{m\,yr^{-1}}$ to $40\,\mathrm{m\,yr^{-1}}$ as opposed to $0\,\mathrm{m\,yr^{-1}}$ which is the same initial meltrate value for all runs of the forward branch). The negligible occurrence of backward R-tipping in our simulations can be explained by the fact that prior to the tipping of the ice sheet towards its advanced state almost no bed response takes

place (Fig. 4c), i.e., a fundamental requirement for R-tipping is lacking here.

To investigate the sensitivity of our results to the steepness of the retrograde bed section, which plays a major role in the MISI mechanism, all simulations were carried out for a shallow ($D_{\mathrm{BD}} = 300$ m) and a deep initial overdeepening ($D_{\mathrm{BD}} = 900$ m), respectively (Fig. 2). This way, large-scale characteristics of the retrograde bed slopes of different Antarctic regions are reflected in our study. For the shallow overdeepening the initial average magnitude of the retrograde slope of $1\cdot10^{-3}$ (1 m elevation

change per km) resembles the more gentle bed slopes of the continental shelves below Antarctica's Ross and Filchner-Ronne ice shelves (Kingslake et al., 2018, see their Extended Data Fig. 3). In the case of the deep overdeepening, the average retrograde bed slope is steeper by a factor of three ($3\cdot10^{-3}$) and is comparable to the observed magnitude of bed drop along a transect through Thwaites Glacier (bed drop of about $750$ m over the first $250$ km upstream of its GL; Morlighem et al., 2020; Sergienko and Wingham, 2022). While a steeper retrograde bed slope generally implies a lower absolute B-tipping threshold, $m_{\mathrm{B-tip}}$,

in our simulations, relevant differences in B-tipping threshold increase relative to the fixed-bed case, $\Delta m_{\mathrm{B-tip}}$, only occur for the thinnest lithosphere applied (Fig. S4a,b). In this case, our model setup of steep (Thwaites-like) retrograde slope yields $\Delta m_{\mathrm{B-tip}}$ values that are roughly halved compared to our setup of a three times shallower retrograde slope. This suggests that WA-type bed characteristics are particularly effective in delaying or even preventing (unstable) ice-sheet retreat in regions of shallow retrograde bed slopes. Ranges of R-tipping are generally slightly larger in the case of the shallow retrograde slope

(Figs. 5 and S3).





All of the simulations exhibiting R-tipping show a GL overshoot in the aftermath of ice-sheet collapse, i.e., a temporary retreat of the GL beyond its ultimate (further downstream) equilibrium position. The magnitude of the GL overshoot is more pronounced for a (1) higher upper-mantle viscosity due to the larger lag between ice-mass change and bed response, (2) thinner lithosphere due to the more localized bed uplift promoting GL re-advance and (3) shallower pro-grade bed slope of the inland sill that eases both GL retreat and subsequent re-advance. Kingslake et al. (2018) provide paleo evidence for extensive overshoots of West Antarctica's GLs since the Last Glacial Maximum, based on the same physical mechanism: the delayed bed rebound in response to large-scale ice-sheet retreat induces GL re-advance. Kingslake et al. (2018) suggest that these overshoots took place on a scale of several hundred kilometers in Antarctica's Ross and Weddell Sea Sectors, partly facilitated by the uplift of regional ice rises that enhanced ice-shelf buttressing. They modeled such GL overshoots in paleo simulations using PISM with the same ELVA model as used here, prescribing upper-mantle viscosities of $\eta \geq 5 \cdot 10^{20}$ Pa s (lithosphere thickness ranging from $T_e = 40$ to 90 km). In our simulations, GL overshoots occur for a similar range of solid-Earth parameter configurations ($\eta \geq 5 \cdot 10^{19}$ Pa s and $T_e \leq 140$ km) but the overshoot size is smaller in most instances (ranging from several 10 km up to 150 km), numbers which are not only determined by the three conditions mentioned above but also by the length scale of our topographic setup.

Conditions that produce sufficiently large GL overshoots in our simulations lead to self-sustaining oscillations between ice-sheet collapse and re-advance (Figs. 9, S3o,p and Movie S2). The oscillations are self-sustaining since – once a critical forcing threshold is crossed – they continue perpetually under constant external forcing solely due to the internal interaction between ice-sheet dynamics and solid-Earth. The oscillations show characteristics of Heinrich events, i.e., periodic phases of abrupt ice discharge of the Laurentide Ice Sheet which are suggested to have occurred during the last glacial period (Heinrich, 1988). The mechanism underlying our simulated oscillations between retreated and re-grown ice-sheet states is internally-driven, similar to the "binge-purge" mechanism described by MacAyeal (1993), and in contrast to several external, climate-forcing driven mechanisms (Hulbe et al., 2004; Alvarez-Solas et al., 2013; Bassis et al., 2017), suggested to explain the periodic occurrence of Heinrich events. While the binge-purge theory is based on thermodynamics at the ice-bed interface (neglected here), the mechanism in our simulations relies on the purely mechanical interaction between ice-sheet dynamics and the deforming solid Earth. A theory proposed by Bassis et al. (2017) also involved the process of bed deformation though not as the direct cause of reversing collapses/advanced ice-sheet states (as in our study) but acting as a control to the external ocean forcing. Simulations by Zeitz et al. (2022) showed large-scale oscillations of Greenland's ice volume resulting from the interaction between ice-sheet dynamics, bed deformation and elevation-dependent surface melting and precipitation. We emphasize that the very simple mechanism of cyclic ice-sheet collapse and re-growth described here should not explain the occurrence of Heinrich events but rather highlight that under certain conditions the basic interaction between ice sheet and solid Earth alone can lead to large-scale, cyclic events of ice-sheet collapse and re-growth that resemble the abrupt characteristics of Heinrich events.

The rates of the slow forcing ramp-up/down prescribed in the hysteresis experiments are chosen such that the response of the coupled system is generally in quasi-equilibrium with the forcing. That is, the very slow forcing rate induces a very slow



response of the ice discharge (ice-load change) so that the associated solid-Earth response remains close to equilibrium with respect to the ice-load change. However, a perfect equilibrium response would be achieved only for an infinitely slow forcing rate, which is possible only in theory. As a result, the B-tipping thresholds inferred here are likely biased towards lower values as the application of even slower forcing rates would allow for a more complete response of the solid Earth (more uplift) to each increment of GL retreat. This particularly applies to the highest viscosity value prescribed in our ensemble ($\eta = 5 \cdot 10^{22}$ Pa s),

associated with a timescale on the order of $100,000$ yr, compared to a timescale on the order of $1,000$ to $10,000$ yr for the next-lower viscosity value ($\eta = 10^{21}$ Pa s; see Table 2 and Fig. A). For the highest viscosity value we thus reduced the forcing rate by one order of magnitude (forcing rate $\dot{m} = 10^{-6}$ m yr$^{-2}$ equals a meltrate increase of $0.1$ m yr$^{-1}$ per $100,000$ yr) compared to the rest of the ensemble ($\dot{m} = 10^{-5}$ m yr$^{-2}$ equals a meltrate increase of $0.1$ m yr$^{-1}$ per $10,000$ yr).

Note that for simplicity we prescribe a fixed bed topography during the ice-sheet spinup, such that the subsequent ramp-up

experiments start from the same ice-sheet state and the same bed topography (for a given $D_{\mathrm{BD}}$ value). This approach involves the assumption that the ice-sheet geometry is in equilibrium with the same underlying bed geometry, regardless of the specific solid-Earth structure. Naturally, both the ice and bed components would freely evolve into equilibrium in an interactive way, resulting in a specific steady-state shape of the bed depression for each solid-Earth configuration. While this approach would have the advantage of delivering a variety of self-consistent equilibrium ice-bed configurations it would substantially limit the

comparability of the coupled ice-bed evolution between different model configurations which we strive for in this study.

The ELVA model applied in this study to model the visco-elastic solid-Earth response accounts for the effect of bed deformation but neglects gravitational and rotational sea-level effects as well as lateral variations in the mantle viscosity and elastic lithosphere thickness. Taking into account the gravitational pull of the ice masses and the associated effect on the regional sea-surface height would likely enhance the stabilizing GIA feedback shown in Fig. 1: ice-mass loss (gain) in the course of ice-sheet

retreat (advance) would immediately reduce (increase) the pull on the regional water masses and thus decrease (increase) the relative sea level regionally, hampering GL retreat (advance). The incorporation of this effect would thus likely lead to higher (lower) thresholds of ice-sheet collapse (re-advance) throughout our ensemble of simulations. Rotational effects as a result of mass redistribution on the sphere produce rather far-field sea-level patterns, and can be neglected for the regional processes considered in our experiments (Gomez et al., 2018). In reality, we find a laterally and vertically varying 3D Earth structure,

such that the local visco-elastic response is also affected by the efficiency of the viscous material flow through the mantle from or to neighboring regions. For our experiments with homogeneous low mantle viscosity and hence efficient material transport we evaluate the inferred rate and magnitude of the visco-elastic deformational response as an upper bound.

Despite the limitations of the applied simplified GIA model discussed above, our results account for the leading-order effect of the ice-bed interaction in the course of MISI-type retreat or advance and allow for a computationally efficient application

in ensemble simulations. A comprehensive investigation of combined gravitational, rotational and deformational effects would require the application of a more complex and computationally demanding 3D GIA and sea-level model (e.g., Albrecht et al.,



2024) instead of the rather simple and fast 1D ELVA model used here (see Swierczek-Jereczek et al., 2024, for an overview of commonly used GIA model classes).

Our model setup involves further simplifying assumptions that affect the results: the accumulation rate at the ice surface, the friction coefficient at the ice base and the ice softness (ice temperature) each are chosen to be uniform across the entire model domain and constant in time. Processes like ice fracturing, strain heating or melting of the ice at the surface or the grounded ice base are neglected. While this approach is strongly simplified compared to the complexity of the real world, it allows us to focus on the interaction between ice dynamics and the evolving solid Earth, avoiding potential interference with effects from other processes that would complicate the analysis and dilute the focus of this study. A comprehensive investigation of the tipping dynamics of the Greenland and Antarctic ice sheets and other important tipping elements of the Earth system that accounts for the complex nature of each element and also their potential interaction, using realistic model setups and involving state-of-the-art Earth System Models, is the aim of the Tipping Points Modelling Intercomparison Project (TIPMIP, Winkelmann et al., 2025).

While the friction coefficient entering the Weertman-type friction law used here is prescribed to be spatially and temporally uniform, the basal stresses calculated from the friction law are not as they depend on the basal ice velocities which, in turn, are diagnosed from the non-local SSA stress balance governing the ice flow. Since the basal stresses are integral part of solving the SSA stress balance which also involves the gravitational driving stress and horizontal membrane and shear stresses, the stress fields adjust over time, in particular allowing for the formation of the ice stream inside the bed trough of the setup (Fig. S1).

Taking into account thermo-mechanically coupled flow in our simulations, would involve strain heating and generation of meltwater at the ice base due to frictional heating (Feldmann and Levermann, 2017), generally leading to a higher sensitivity of the ice flow to perturbations in our simulations, strengthening the MISI feedback and thus likely result in overall lower tipping thresholds.

We use a spatially uniform sub-ice-shelf meltrate as the single control parameter to alter ice-shelf buttressing in our hysteresis experiments which is observed to be the main mechanism driving Antarctic ice-sheet retreat. Applying a realistic, spatially heterogenous meltrate pattern would likely alter our results. According to observations, basal meltrates are typically strongest close to the GL beneath the thickest portion of the ice shelf (e.g., Dutrieux et al., 2013; Shean et al., 2019; Adusumilli et al., 2020). In some places, observations show concentrated melting at the lateral ice-shelf shear margins (Berger et al., 2017; Shean et al., 2019; Alley et al., 2019) which would accelerate buttressing loss (Feldmann et al., 2022), strengthening the MISI feedback in our simulations and thus lead to lower tipping thresholds.

## 5 Conclusions

In summary, our results highlight how crucially the type of the solid-Earth structure underlying a MISI-prone ice sheet can affect its tipping dynamics in a qualitative and quantitative way, based on the tightly coupled evolution of ice sheet and bed





and the involved (de-)stabilizing feedbacks: weak (West-Antarctic type) solid-Earth structures efficiently counteract MISI initiation due to a very fast and localized bed uplift that fuels a strong stabilizing GIA feedback. Conversely, strong (East-Antarctic type) solid-Earth structures, responding comparatively slow to ice-mass changes, can substantially reduce critical tipping thresholds for a sufficiently fast forcing rate, i.e., lead to the occurrence of R-tipping. Furthermore, a more localized and slower solid-Earth response promotes GL overshoot, which, in case of sufficiently shallow bed slopes, can result in the occurrence of self-sustaining oscillations between advanced and collapsed ice-sheet states, induced solely by the dynamic interaction between ice sheet and solid Earth. Regarding projections of the future stability of the Antarctic Ice Sheet and its associated sea-level contribution, our results in particular suggest that considering both the *magnitude* and the *rate* of future anthropogenic greenhouse gas emissions driving global warming as well as a careful spatial and temporal representation of Antarctic bed deformation with associated uncertainties are key for making insightful statements.



*Code and data availability.* The model code used in this study is based on PISM stable version 1.0 and can be obtained from https://doi.org/10.5281/zenodo.6531439 (Feldmann, 2022). Model output and scripts to reproduce the simulations and figures of this paper
will be made available through a separate Zenodo archive.

*Video supplement.* Two supplementary videos (Movies S1 and S2) are part of this paper and can be accessed from https://doi.org/10.5446/s_1984. The movie captions can be found in the supplement.

*Author contributions.* All co-authors contributed to the design of the study. JF prepared the experiments, carried out the numerical simulations and analyzed the model output. JF discussed the results with AKK and TA. JF wrote the paper with contributions from AKK and
555 TA.

*Competing interests.* The contact author has declared that none of the authors has any competing interests.

*Acknowledgements.* This research was supported by OCEAN ICE, which is co-funded by the European Union, Horizon Europe Funding Programme for research and innovation under grant agreement Nr. 101060452 and by UK Research and Innovation. OCEAN ICE Contribution number XX.




**Table 1.** Physical constants and parameter values as prescribed in the simulations

| Parameter | Value | Unit | Physical meaning |
|---|---|---|---|
| $a$ | 0.5 | $\mathrm{m\,yr^{-1}}$ | Surface accumulation rate |
| $A$ | $3.169 \cdot 10^{-25}$ | $\mathrm{Pa^{-3}\,s^{-1}}$ | Ice softness, related to ice temperature $T$ |
| | | | via Arrhenius law (Glen, 1955) |
| $T$ | $-13.2$ | °C | Ice temperature |
| $C$ | $3.981 \cdot 10^{6}$ | $\mathrm{Pa\,m^{-1/3}\,s^{1/3}}$ | Basal friction parameter, |
| | | | entering Eq. (3) of Cornford et al. (2020) |
| $E$ | $8.2 \cdot 10^{10}$ | $\mathrm{N\,m^{-2}}$ | Young's modulus |
| $\nu$ | 0.25 | | Poisson's ratio |
| $g$ | 9.81 | $\mathrm{m\,s^{-2}}$ | Gravitational acceleration |
| $m$ | 1/3 | | Basal friction exponent, |
| | | | entering Eq. (3) of Cornford et al. (2020) |
| $n$ | 3 | | Exponent in Glen's law |
| $\rho_i$ | 918 | $\mathrm{kg\,m^{-3}}$ | Density of ice |
| $\rho_o$ | 1028 | $\mathrm{kg\,m^{-3}}$ | Density of ocean water |
| $d_c$ | 500 | m | Initial depth of bed trough compared with side walls, |
| | | | entering Eq. (1) of Cornford et al. (2020) |
| $f_c$ | 4 | km | Characteristic width of bed-trough side walls, |
| | | | entering Eq. (1) of Cornford et al. (2020) |
| $w_c$ | 160 | km | Half-width of bed trough, |
| | | | entering Eq. (1) of Cornford et al. (2020) |
| $D_{\mathrm{BD}}$ | 300, 900 | m | Depth of bed depression below sea level |
| $L_x$ | 800 | km | Length of computational domain ($x$ dimension) |
| $L_y$ | 320 | km | Width of computational domain ($y$ dimension) |
| $x_{\mathrm{CF}}$ | 780 | km | Position of fixed calving front |
| $\eta$ | $\{10^{18}, 5 \cdot 10^{19}, 10^{21}, 5 \cdot 10^{22}\}$ | $\mathrm{Pa\,s}$ | Upper-mantle viscosity |
| $T_e$ | $\{20, 80, 140, 200\}$ | km | Lithosphere thickness |





**Table 2.** Scaling of response times of the solid Earth, $\tau_{\mathrm{SE}}$, for the upper-mantle viscosities, $\eta$, used in the experiments. The second column gives the scaling expected from theory (see Appendix A), matching with the absolute timescales derived from perturbation experiments (see Fig. A1). For comparison, the absolute response time of the ice load in the absence of bed deformation, is given for the stable regime, $\tau_{\mathrm{L,st}}$, (prior MISI initiation; see Fig. A2) and for the unstable regime (during MISI), $\tau_{\mathrm{L,MISI}}$.

| $\eta$ (Pa s) | $\tau_{\mathrm{SE}}$ (theory) | $\tau_{\mathrm{SE}}$ (experiments) | $\tau_{\mathrm{L,st}}$ | $\tau_{\mathrm{L,MISI}}$ |
|---|---|---|---|---|
| Fixed bed | - | - | $\approx 6{,}000$ a | $\sim 10-100$ a |
| $1 \cdot 10^{18}$ | $\tau_{\mathrm{ref,th}}$ | $\tau_{\mathrm{ref,ex}} = 7.5$ a | - | - |
| $5 \cdot 10^{19}$ | $50 \cdot \tau_{\mathrm{ref,th}}$ | $350$ a | - | - |
| $1 \cdot 10^{21}$ | $1{,}000 \cdot \tau_{\mathrm{ref,th}}$ | $7{,}000$ a | - | - |
| $5 \cdot 10^{22}$ | $50{,}000 \cdot \tau_{\mathrm{ref,th}}$ | $350{,}000$ a | - | - |





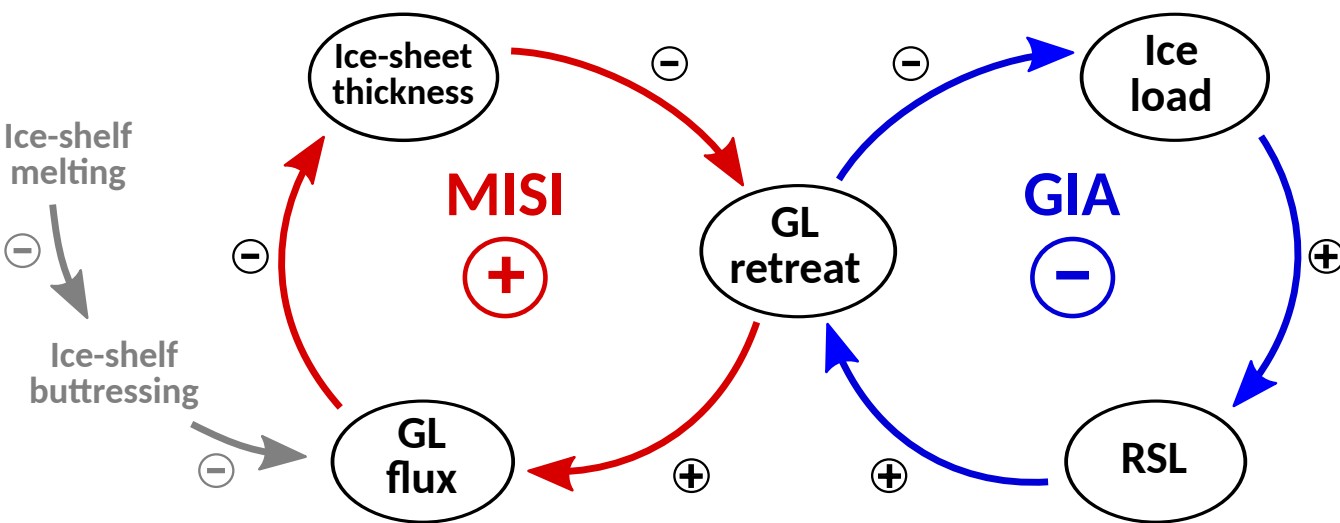

**Figure 1.** The two main feedback mechanisms, governing the dynamics of the coupled ice-sheet-solid-Earth system. Triggered by basal ice-shelf melting (grey), the self-enforcing MISI feedback (red) promotes GL retreat on retrograde bed and leads to ice-sheet collapse if ice-shelf buttressing is weak enough. Conversely, the dampening GIA feedback (blue) hampers GL retreat and thus counteracts ice-sheet collapse. RSL stands for relative sea level.



**Figure 2. (a)** Bed topography (with deep bed depression, $D_{BD} = 900$ m) prescribed during model spinup with resulting steady-state GL position and fixed calving front (grey contours). **(b)** Centerline (dotted line in panel a) profiles of bed and ice-sheet-shelf system for different snapshots of a hysteresis simulation (slow ramp-up of basal melt).



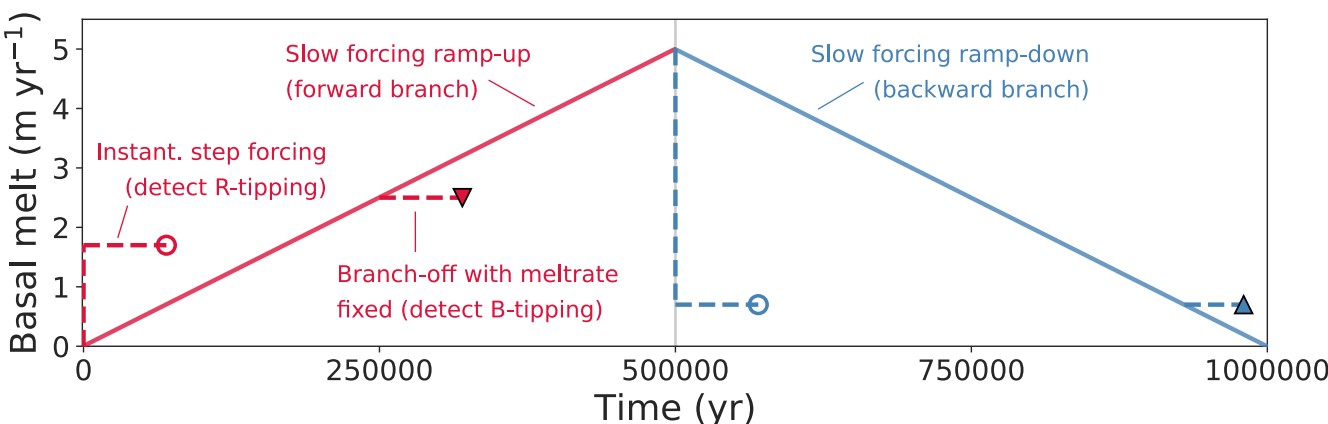

**Figure 3.** Exemplary forcing timeseries. Slow forcing ramp-up (red) and ramp-down (blue) applied in the hysteresis experiments (Sect. 2.2.4). Branch-off experiments (Sect. 2.2.5), from the ramp-up/ramp-down with the current meltrate held fixed to simulate the associated equilibrium state and infer B-tipping thresholds, are indicated by triangles. Step-forcing experiments (Sect. 2.2.6) that apply an infinitely fast forcing ramp-up/ramp-down to a specific meltrate value in order to check for possible rate-dependency of the resulting steady state to the forcing rate are highlighted by a circle. The resulting hysteresis curve with inferred B-tipping thresholds and R-tipping range are shown in Fig. 4.







**Figure 4. (a)** Hysteresis curves of the centerline GL position for slow ramp-up (red) and subsequent ramp-down (blue) of basal ice-shelf melt forcing ($1 \,\mathrm{m\,yr^{-1}}$ per $100,000$ model years) with solid-Earth parameters $\eta = 10^{21} \,\mathrm{Pa\,s}$ and $T_e = 80 \,\mathrm{km}$ and an initial bed depression depth of $D_{\mathrm{BD}} = 900 \,\mathrm{m}$. Red and blue areas give the ranges of B-tipping. Light red area highlights the range of R-tipping. Triangles represent steady states after branching off from the hysteresis curve at fixed melt rate. Circles denote equilibrium states after an instantaneous step forcing to the respective melt rate (see Fig. 3 for the different forcings). Grey curves correspond to the case of a fixed bed. **(b)** Evolution of the centerline bed topography (1,000-year snapshots) during forcing ramp-up with GL position marked by small circles. Dashed contours represent ice-shelf draft and bed profile for initial and final equilibrium state (grey) and the last stable state before collapse (black). **(c)** Evolution of GL and bed during forcing ramp-down, analogous to panel b. The B-tipping thresholds are marked by dashed lines in the colorbars.





**Figure 5.** Hysteresis curves of the centerline GL position. Each panel represents one combination of the two solid-Earth parameters $\eta$ (increasing from left to right) and $T_e$ (increasing from bottom to top). Red and blue curves show the quasi-equilibrium trajectories for the slow forcing ramp-up and ramp-down, respectively. Grey curves show the response for a fixed bed. Dashed vertical lines indicate the inferred B-tipping thresholds. Note the extended range of the $x$ axis (basal meltrate) for $T_e = 20$ km due to the occurrence of comparatively large B-tipping thresholds.



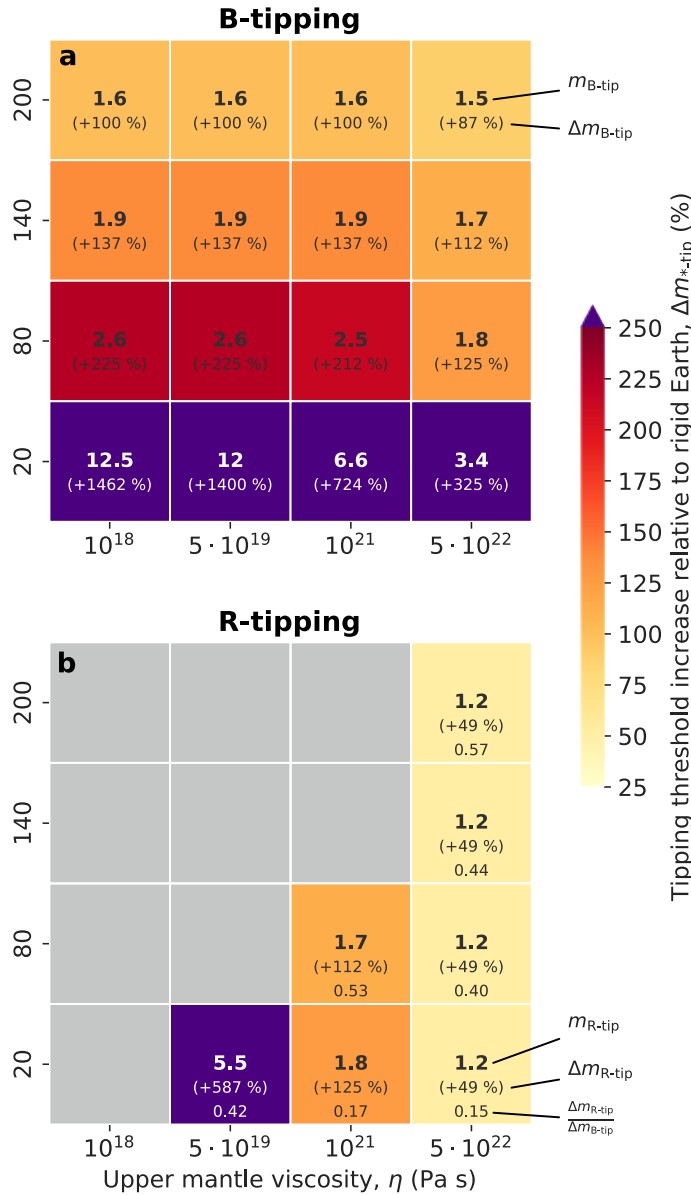

**Figure 6.** Heat maps showing the shift in the critical meltrate threshold relative to the fixed-bed case for **(a)** B-tipping and **(b)** potential R-tipping. The reference tipping threshold of the fixed-bed case is $m_{\mathrm{B-tip,fixed}} = 0.8 \, \mathrm{m \, yr^{-1}}$. The absolute values of the respective tipping thresholds, $m_{\mathrm{B-tip}}$ and $m_{\mathrm{R-tip}}$, are given by bold large numbers and the values of tipping threshold increase relative to the fixed-bed case, $\Delta m_{\mathrm{B-tip}}$ and $\Delta m_{\mathrm{R-tip}}$, are given in brackets (see annotations in panels a and b and Eqs. 1 and 2). The third row in each tile of panel b states the ratio between $\Delta m_{\mathrm{R-tip}}$ and $\Delta m_{\mathrm{B-tip}}$, indicating the strength of the reduction of the critical tipping threshold in case of R-tipping (values closer to zero associated with stronger reduction).





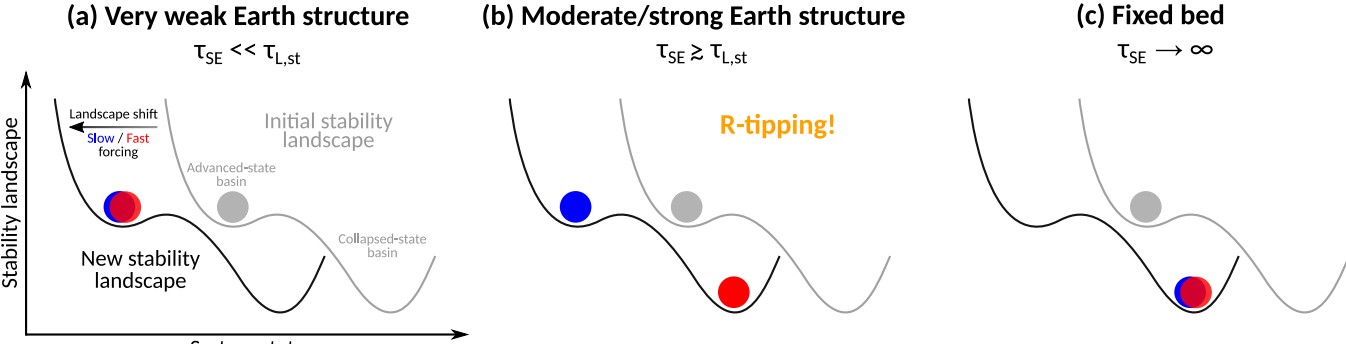

**Figure 7.** Schematic to illustrate under which conditions R-tipping emerges in our simulations (inspired by Ritchie et al., 2023, Fig. 1). In each panel the grey ball represents the initial steady state of the ice sheet (associated with the grey initial stability landscape) which is perturbed by basal ice-shelf melting (ramp-up forcing that shifts the stability landscape to a new state highlighted in black) with resulting final steady states represented by colored balls (blue for a slow forcing and red for a fast forcing). **(a)** If the Earth structure is very weak, then the solid-Earth response is quasi-instantaneous and thus, independently of the forcing rate (slow or fast shift of the stability landscape), the system remains in its advanced stable state. **(b)** For a moderate to strong Earth structure, if the timescale of ice-load change is similar to or faster than the timescale of the solid-Earth response, then the response of the system depends on the forcing rate, i.e., it collapses (remains stable) for a fast (slow) shift of the stability landscape. **(c)** If the bed elevation is held fixed, then the timescale of the solid-Earth response is infinite and thus the ice sheet will tip into a collapsed state, independently of the forcing rate. All cases assume that the forcing is ramped up to a value that is (1) below the critical B-tipping threshold that is specific to the applied solid-Earth structure, $m_{\mathrm{B-tip}}$, and (2) above the B-tipping threshold of the fixed-bed case, $m_{\mathrm{B-tip,fixed}}$.





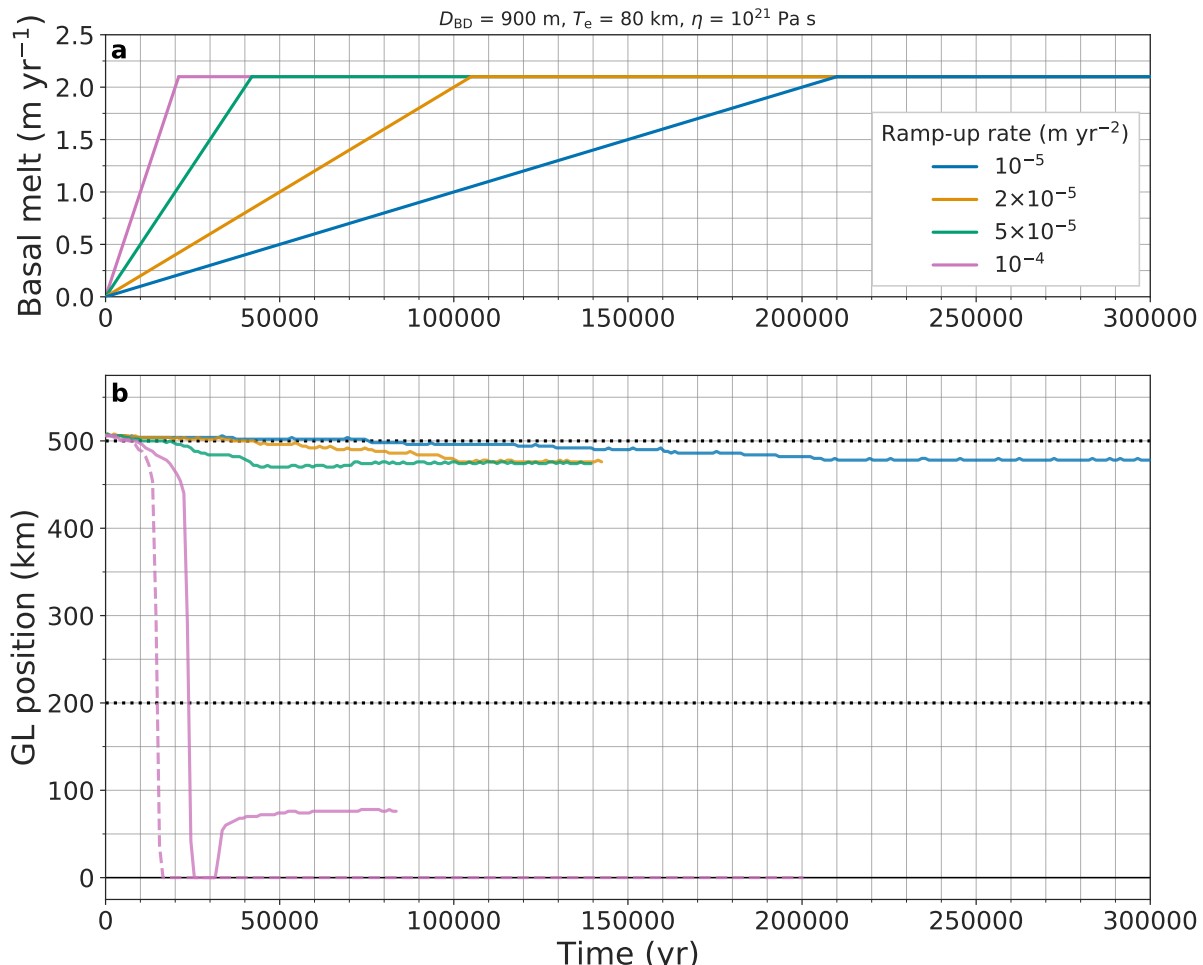

**Figure 8.** Timeseries of **(a)** different ramp-up rates of applied sub-ice-shelf melt forcing and **(b)** associated response of the centerline GL position, highlighting the rate-dependency of a possible ice-sheet destabilization. Dashed line represents the case of a fixed bed. The two dotted horizontal lines mark the range of retrograde bed slope.





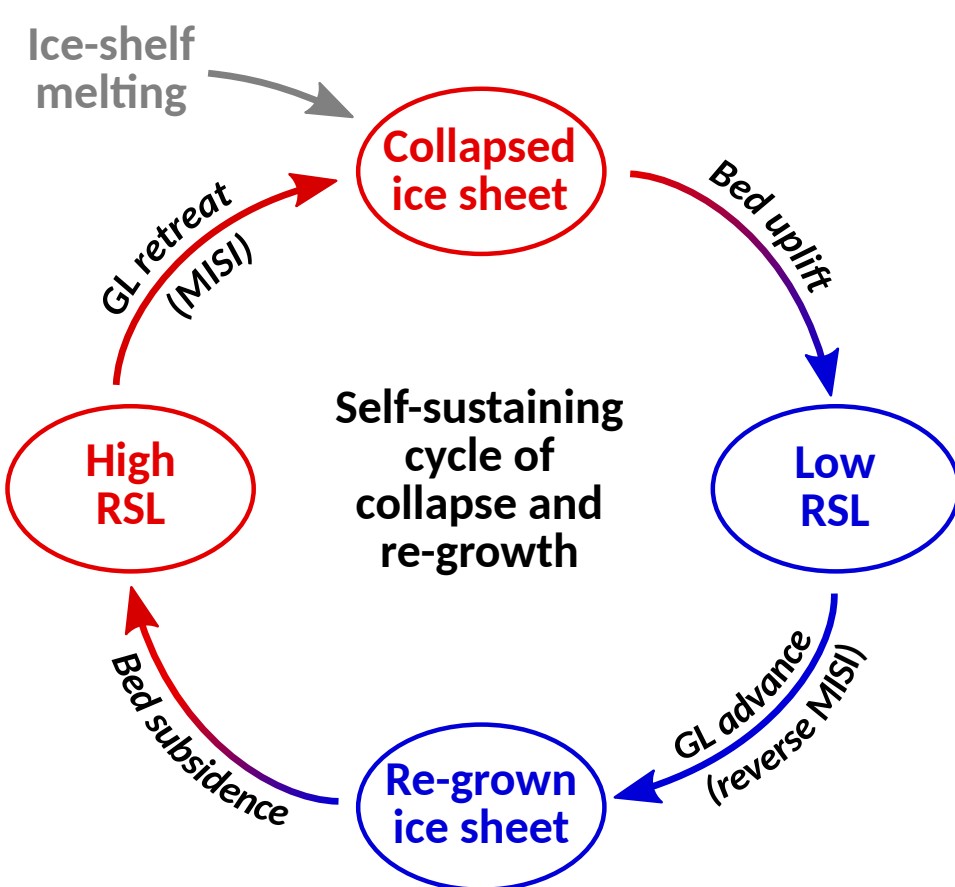

**Figure 9.** Cycle of processes underlying the self-sustaining oscillations between advanced and collapsed ice-sheet states.



**Appendix A: Timescales**

**A1 Maxwell times**

Due to the assumption of a linear-viscous Maxwell rheology made in the LC model, the timescale of the viscous response of the solid Earth to ice-load changes, named $\tau_{\mathrm{SE}}$ hereafter, scales linearly with the mantle viscosity. That is, for two different mantle viscosities $\eta$ and $\eta'$ and the associated timescales $\tau_{\mathrm{V}}(\eta)$ and $\tau_{\mathrm{V}}(\eta')$, it holds

$$\frac{\tau_{\mathrm{V}}(\eta)}{\tau_{\mathrm{V}}(\eta')} = \frac{\eta}{\eta'}, \tag{A1}$$

giving *relative* timescales of the solid Earth response. To infer the *absolute* timescales which are intrinsic to our model configurations, we carried out a small set of additional perturbation experiments, modeling only the viscous response of the bed: starting from the initial equilibrium, the entire ice load is removed instantaneously, inducing abrupt bed uplift. The bed elevation converges to a new equilibrium on a viscosity-specific timescale, which we infer by an exponential fit to the curve of bed-elevation change (Fig. A1). For the lowest applied mantle viscosity, $\eta_{\mathrm{min}} = 10^{18}\ \mathrm{Pa\,s}$, the fit yields $\tau_{\mathrm{SE}}(\eta_{\mathrm{min}}) \approx 7.5\ \mathrm{yr}$. Based on this value, the linear scaling according to Eq. (A1) then results in fitted curves that match the simulated curves for each of the higher mantle viscosities to a good approximation (summarized in Table 2).

The critical ramp-up timescales that lead to R-tipping, $\tau_{\mathrm{R}}$, also obey the linear scaling of Eq. (A1). For a rough estimate, we compare the lowest forcing rates $r$ (ramp-up timescales $\tau \sim r^{-1}$) at which tipping occurs for two different upper-mantle viscosities (taken from Figs. 8 and S5).

$$\frac{\tau_{\mathrm{R}}}{\tau_{\mathrm{R}}'} = \frac{r'}{r} = \frac{2 \cdot 10^{-3}\ \mathrm{m\,yr^{-2}}}{10^{-4}\ \mathrm{m\,yr^{-2}}} = 20 = \frac{10^{21}\ \mathrm{Pa\,s}}{5 \cdot 10^{19}\ \mathrm{Pa\,s}} = \frac{\eta}{\eta'}. \tag{A2}$$

**A2 Timescale of stable GL retreat**

Besides $\tau_{\mathrm{SE}}$, the other essential timescale in our model configuration is the response time of GL retreat and associated ice-load reduction to a perturbation of basal ice-shelf melting in the absence of bed deformation. For GL retreat in the stable regime, i.e., retreat on the shallow retrograde slope close to the tip of coastal sill, it is on the order of several thousand years ($\tau_{\mathrm{L,st}} \sim 6000\ \mathrm{yr}$). This value is inferred for the GL response to a basal-melt step forcing from zero to $1\ \mathrm{m\,yr^{-1}}$ (for $D_{\mathrm{BD}} = 300\ \mathrm{m}$) and $0.7\ \mathrm{m\,yr^{-1}}$ (for $D_{\mathrm{BD}} = 900\ \mathrm{m}$), respectively (Fig. A2). The forcing to values just below the B-tipping thresholds ensures perturbations that are large enough to induce GL retreat while still sufficiently small to avoid triggering a MISI.




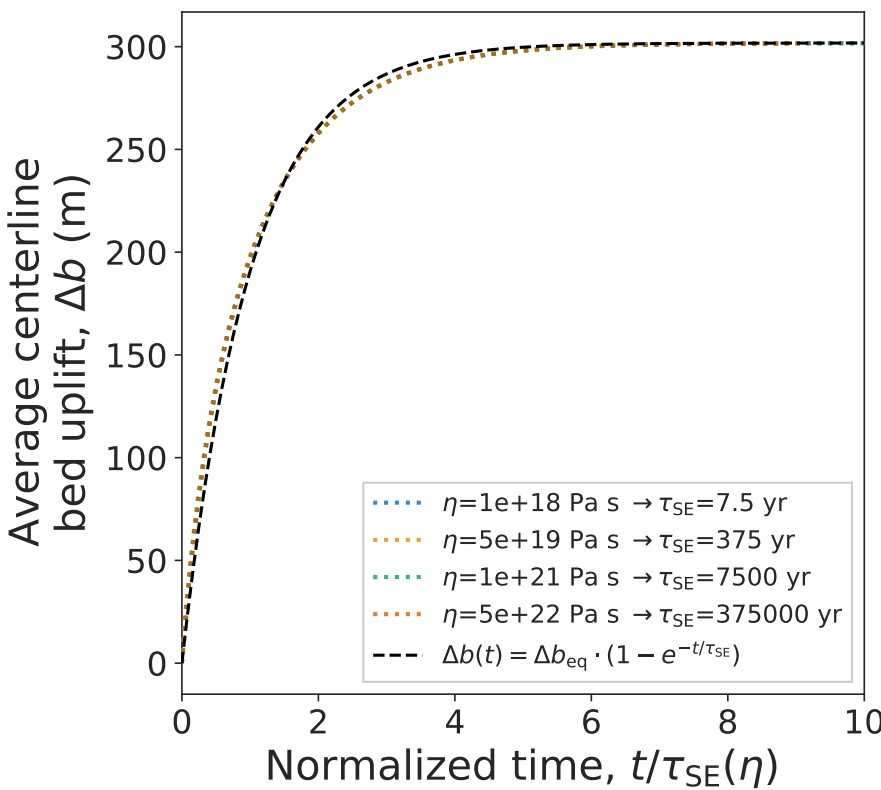

**Figure A1.** Viscous bed response to an instantaneous removal of the entire ice load for all four upper-mantle viscosities used in this study. Here we choose the bed response, $\Delta b$, to be represented by the spatial average of the bed elevation change (uplift) along the centerline of the setup compared to the unperturbed initial steady state. The black dashed line represents the exponential fit according to the function given in the legend with viscosity-specific response timescales $\tau_{SE}$. Note that for each viscosity value the time axis is normalized to the respective response timescale given in the legend. Since all curves align and are well approximated by the exponential fit the experimental results obey the theoretically expected timescaling. The final steady-state value of the uplift anomaly is $\Delta b_{eq} = 302$ m in all cases. The simulations use a lithosphere thickness of $T_e = 80$ km and an initial bed-depression depth of $D_{BD} = 300$ m.





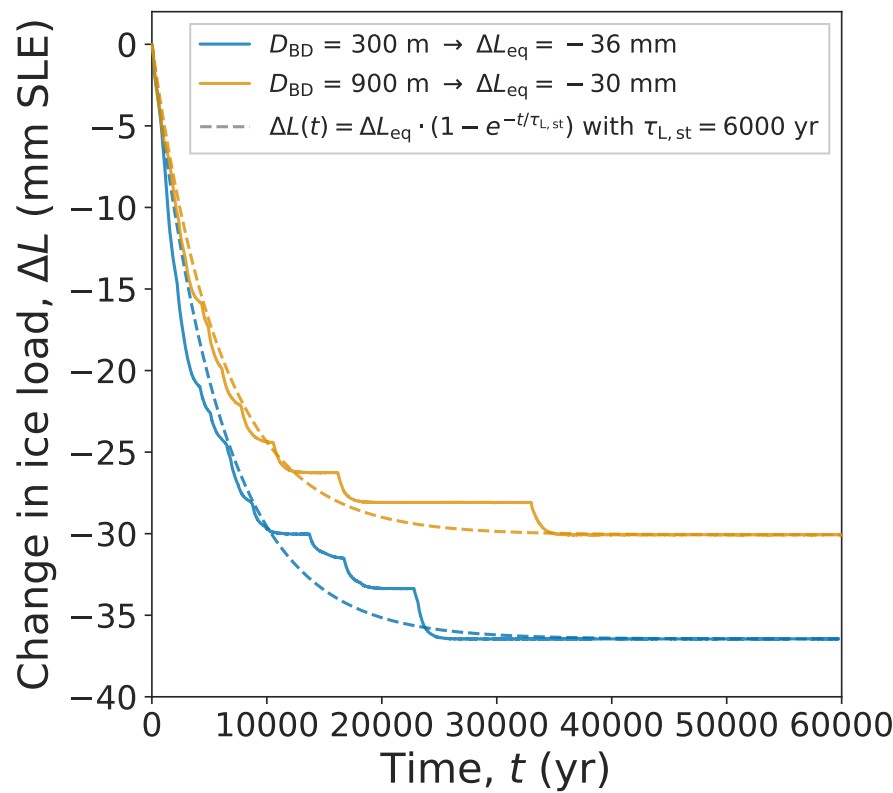

**Figure A2.** Change in ice load, $\Delta L$, with respect to the initial steady state and associated timescale, $\tau_L$ (relaxation time of the ice load), to a basal-melt step forcing of $0.7 \, \mathrm{m \, yr^{-1}}$ that leaves the ice sheet in the stable regime for both bed-depression depths. The dashed lines show the exponential fits according to the equation given in the legend using the respective final steady-state value of ice-load change, $\Delta L_{eq}$, and a response time of $\tau_L = 6000$ a.



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
