# Peer review of "Rate-induced tipping of ice sheets due to visco-elastic Earth response under idealized conditions"

_EGUsphere, 2025_

## Referee Comment (RC1)

**General comments**

The authors investigate the role of vertical bedrock displacement on the marine ice-sheet instability. To this end, they sample various combinations of the lithospheric thickness and upper-mantle viscosity, and use an idealised setup that mimics some of the important features of Thwaites Glacier, which is likely to undergo a marine ice sheet instability in the coming centuries. By forcing the system with a very slowly evolving meltrate from the ocean, the authors assess the bifurcation point of the system. By using a forcing with significant rate, they simulate a collapse of the glacier below the bifurcation point, a situation also known as R-tipping. For some parameter combination, they observe a back and forth migration of the grounding line position across the unstable region at a constant forcing. This results in three main messages of the paper: varying the lithospheric thickness and upper-mantle viscosity can (1) shift the bifurcation point, (2) lead to R-tipping and (3) give rise to self-sustained oscillations.

The main messages conveyed by the authors are undoubtedly relevant for past and future marine ice-sheet evolution. The setup of lower complexity eases the understanding of the mechanisms at play and facilitates a clear description and visualization of the results. The text is easy to follow, the figures are nice and informative. It is obvious that the authors put great effort into the present work and I am sure that it will be a valuable contribution for the community.

However, I believe that a few important points need to be checked/modified before a possible publication.

The authors find that the lithospheric thickness, $T_e$, plays a significant role in shifting the B- and R-tipping point. The explanation for this is found in Fig. S2, where it is shown that the uplifted bathymetry is very different for each choice of $T_e$. On many occasions, the authors mention that smaller values of $T_e$ lead to a more localized response, but it really looks like the decisive factor is that the response is emphatically larger. For instance, the maximal bedrock uplift yields nearly 1,000 m for $T_e = 20$ km but only about 300 m for $T_e = 200$ km. Besides the fact that the authors should emphasize this more and move Fig. S2 to the main text, I am very surprised by this result: other studies have shown that, although $T_e$ affects the gradient and rate of the displacement, it only marginally affects its equilibrium magnitude (Whitehouse et al., 2012; Milne et al., 2018; Coulon et al., 2021; Swierczek-Jereczek et al., 2024). I propose a simple way to verify this in the specific comments.

However, I believe that the strong dependence on $T_e$ cannot be solely explained by the differences in uplift and the authors do not dwell into the details of this, which is a shortcoming that needs to be addressed. I suggest the following explanation. The bathymetry of the side walls is close to 0 around the coastal sill. Thus, any significant bed uplift suppresses basal melt there and provides lateral pinning points that greatly hinder the grounding-line retreat. Therefore, the role of $T_e$ in shifting the bifurcation point so strongly might be largely due to the choice of the bathymetry. Given the large amount of work already provided by the authors, I don't think they should perform additional runs to investigate this. However, they should show heatmaps of the fully uplifted bathymetry (with a colour bar that

changes rapidly around z=0, for various T_e, optionally in supplement, not only transects of the centreline) and critically discuss the fact that their setup is particularly sensitive to bedrock uplift because of the large pinning-point potential, right before the retrograde portion of the bedrock.

Furthermore, the update frequency of the LC model is set to 10 years, although the authors infer a characteristic timescale of the solid-Earth of 7.5 years for eta = 10^18 Pa s. I think the simulations involving low values of eta need to be performed again with a higher update frequency, unless the authors can prove that this is not critical. Of course, the LC model is unconditionally stable since it uses an implicit time-stepping scheme, but numerical stability does not imply numerical accuracy.

The authors use the case of a rigid Earth as a baseline to quantify R-tipping (L. 15-16, L. 250-258, Eq. 2, Fig. 6) and therefore compare setups with different physics (GIA on vs. GIA off). This is not the idea of R-tipping. However, it is easy to address by comparing simulations with the same parameters (T_e, eta) but different rates, i.e. modifying Eq. 2 to:
\Delta m_R-tip(eta, Te) = (m_R-tip(eta, Te) - m_B-tip(eta, Te)) / m_B-tip(eta, Te) * 100%

I am confident that the authors will be able to address these points, and I wish them the best of luck for the rest of the publication process.

**Specific comments**

1. The title focuses on R-tipping, but the paper also highlights how certain solid-Earth parameter choices can shift the bifurcation point or give rise to self-sustained oscillations. Therefore, I feel like the title could be more general, something like "Solid-Earth structure conditions the stability of marine-terminating glaciers". But this is up to you.
2. Mentioning the large rate of the anthropogenic global warming compared to the past temperature record would be a nice complement for the motivation of studying R-tipping.
3. L. 60-61: As mentioned above, T_e should not affect the vertical displacement significantly in the LC model.
4. L. 69. You say that R-tipping remains largely unstudied in the MISI context, without any citation. However, in the discussion, you cite Swierczek-Jereczek et al. (2025), which has already suggested the timescale separation between ice and GIA to be responsible for R-tipping. This feels odd, especially because you therefore miss a nice motivation of your work in the introduction: to date, R-tipping of the AIS was studied, but never in a setup of lower complexity, where it is easier to identify and isolate the mechanisms. This is an simple but necessary fix to embed your work well in the state of the art.
5. Your model description is really nice and easy to follow, but I feel like the GIA part misses some important information. What is the upper-mantle viscosity? Furthermore, to impose sensible boundary conditions, the LC model requires the ice-sheet domain to be smaller than the GIA domain. What is your choice here? Especially when it comes to the y-dimension, it is not an obvious choice since you assume periodic boundary conditions in the ice-sheet model which are difficult to

consistently treat in the GIA model. From what I see in your plots, the vertical displacement is always very close to zero at x = 800 km, which makes me wonder if you chose the GIA domain to correspond exactly to the ice-sheet domain. Since I am not sure of this, the lines below are speculative.

If you chose your GIA domain to be the same as your ice domain, it would be critical for the experiments with large T_e: since the displacement spreads significantly, it might be that u_viscous << 0 at x = 800 km. However, following Bueler et al. (2007), you would impose u_viscous = 0 there (on average). This might offset your displacement and be responsible for the large changes in displacement magnitude depending on T_e, which what I mentioned in the general comments.

6. To validate the setup with respect to this concern, I would like you to perform a standalone run of the GIA model with T_e = 80 km, assuming a rectangular ice load centred in (x, y) = (0, 0), with L_x = 1200 km, L_y = 320 km, thickness = 1 km at t = 0 and remove the load instantaneously after. Please specify what is your GIA domain and upper-mantle viscosity so that I can verify that we obtain similar results.

7. L. 143: Based on what you say here, I think it would be really nice if you would highlight the plausible solid-Earth parameter combinations in Figs. 5 & 6. This would ease the interpretation.

8. L. 173: I feel like it's a pity to not show your results from branched-off simulations and step experiments in Fig 4.a and Fig. 5 (triangles and circles). This would allow the reader to see the results rather than a post-processed version of it. Please include this.

9. L. 206-208: I understand that what you describe here is just a consequence of the differences in the fully uplifted bathymetries. Based on Fig. S2, it looks like the decisive factor here is whether the fully uplifted centreline bathymetry has any emerged part. Could you confirm/discuss/stress this? This also conditions the possibility of overshoot, which you mention at l. 329, but I think not with sufficient emphasis: the decisive role of GIA here is because it modulates the nonlinearity in basal melt (= 0 if z > 0; > 0 if z<= 0).

10. L. 215: Not sure that the ice shelves are the reason for the readvance: even without buttressing, you observe a similar behaviour, as shown by Schoof (2007); Pattyn et al. (2012).

11. L. 339-340: You should mention that this solid-Earth parameter combination is unlikely.

12. In the discussion, you should mention that MISI can be stabilised by other mechanisms, e.g. upstream increase of precipitation (Sergienko, 2022). This is important, since in the present setup you prescribe constant accumulation and therefore miss this effect.

13. Really neat parallel with binge-purge mechanism and nice link to Holocene readvance!

References:
- Sergienko, 2022: No general stability conditions for marine ice-sheet grounding lines in the presence of feedbacks.
- Whitehouse et al., 2012: A new glacial isostatic adjustment model for Antarctica: calibrated and tested using observations of relative sea-level change and present-day uplift rates: A new GIA model for Antarctica.

- Milne et al., 2018: The influence of lateral Earth structure on glacial isostatic adjustment in Greenland.

**Technical corrections**

i. L. 1: "may be characterized by" → "will likely be subject to"
ii. L. 2-5: This sentence is long, a bit confusing and would benefit from being split into two sentences. "West Antarctica" → "West-Antarctic Ice Sheet"; "most" → "more".
iii. L. 7-8: "basal ice-shelf melting" → "sub-shelf melting"
iv. L. 8: "solid Earth structures" → "solid-Earth structures,"
v. L. 10: "B-tipping threshold" → "bifurcation point"
vi. L. 14: "that for half of the ensemble members rate-induced tipping (R-tipping) occurs" → "that rate-induced tipping (R-tipping) occurs for half of the ensemble members"
vii. L. 16: "effective critical tipping threshold" → "effective tipping threshold"
viii. L. 27: mention a percentage of the WAIS contribution to the AIS ice loss.
ix. L. 34: "van den Akker" is missing a hyperref. This applies to many of your citations throughout the paper, e.g. "Kachuck" (l. 53), "Fürst" (l. 83), "Pattyn" (l.86). This also applies to all the references you are making to figures (e.g. l. 41) and tables (e.g. l. 113). Please check this thoroughly.
x. L. 35-36: cite Swart and Li at the end of the sentence.
xi. L. 38: "on bed" → "on a bed"
xii. L. 41-43: "One important stabilizing factor in the MISI context is the buttressing effect of ice shelves that are laterally confined or grounded on topographic highs" → "The buttressing provided by laterally confined ice shelves is an important stabilizing factor of the MISI".
xiii. You often use the phrasing "MISI-type retreat", as in l. 49. I would simply write "MISI" in most cases, or "MISI-driven retreat". Also at l. 132.
xiv. L. 72: "for" → "of"
xv. L. 117: "(Cornford et al., 2020, Eq. 3 of)" → "(Eq. 3 of Cornford et al., 2020)".
xvi. L. 127: "reaches equilibrium after several 10,000 model years" → just mention the exact number of model years.
xvii. L. 134-135: this is a good sanity check for your model setup but I am not sure if it needs to be mentioned in the paper. Up to you.
xviii. L. 202: "does only occur" → "only occurs"
xix. L. 209: "several 100 m" → "several hundreds of metres"
xx. L. 228: delete "that could counteract GL retreat"
xxi. L. 240: "solid-Earth structure,"
xxii. L. 307: "10km" → "tens of kilometers"
xxiii. Fig. 1: "GL retreat" → "GL retreat on retrograde bed"
xxiv. Fig. 3: Would read better in kyr
xxv. Fig. 9: "reverse MISI" → "MISI". The former sounds a bit odd to me, since it's just MISI, following Schoof (2007).
xxvi. Some references don't include all the required information. For instance, the journal is missing for Adhikari et al. (2014) and some weird information (bandiera_abtest: a Cg_type, Group Subject_term, Subject_term_id…) is given for Adusumilli et al. (2020). Please check the whole reference section thoroughly.

---

## Referee Comment (RC2)

**Reviewer report egusphere-2025-5859**

**Rate-induced tipping of ice sheets due to visco-elastic Earth response under idealized conditions**

January 31, 2026

The present work studies how the interaction between ice-sheet dynamics and the visco-elastic response of the solid Earth affects the tipping dynamics of marine-terminating ice sheets in idealised geometries. A bifurcation point is found via quasi-equilibrium forcing in ocean melt rates. If the forcing rates are increased, a glacier collapse occurs for melt rates below the bifurcation points for certain parameter combinations, thus illustrating R-tipping behaviour. The last message of the study exhibits self-sustained oscillations driven solely by the non-linear interaction between ice flow and solid Earth.

Overall, I find that the message of the paper is successfully conveyed. Idealised geometries are fundamental to isolate mechanism and understand the underlying physical processes. Results presented are clear and well structured, providing valuable conclusions for the scientific community.

Nevertheless, there are a number of major remarks that I would like the authors to address and further elaborate on the manuscript. Particularly, the work falls short in certain aspects that I have listed in the Major remarks section. Moreover, I have included a list with other minor remarks at the end of the document.

**1 Major remarks**

- **Motivation.**

  Line 69 in the introduction reads: "*While the concept of R-tipping has been examined for other tipping elements [...], to date it remains largely unstudied in the MISI context*".

  However, there are previous studies that have already explored the idea that timescale separation between solid-Earth and ice sheets permits R-tipping. For instance, Swierczek-Jereczek et al. (2025): "*Here we propose that sectors subject to MISI can undergo R-tipping: when forcing rates are high and the bedrock uplift is slow, the grounding-zone retreat is not stabilised as efficiently as in equilibrium simulations, triggering MISI at warming levels below the bifurcation point*". Currently, the manuscript under review is motivated by the apparent absence of previous studies where MISI is investigated within the context of R-tipping. Instead, I would suggest framing the motivation in terms of a lack of experiments in idealised geometries where physical mechanisms can be isolated.

- **Spin-up.**

  According to Section 2.2.2, bed deformation is switched off during spin-up and is later justified by: "*[...] enabling bed deformation after the system has equilibrated does not lead to any changes in the ice and bed geometries*". Why is this the case? I guess the GIA model is set up to reflect load changes referring to the ice thickness equilibrated with a fixed bed. I understand that this allows simulations to start from the same grounding line position, but it is only one choice and should be detailed in the text.

  I propose an additional approach to potentially enrich the discussion: performing the spin-up with bed deformation. Equilibrium grounding lines will therefore be different for each combination of $T_e$ and $\eta$. It is quite informative to observe the equilibrated ice sheet profiles with their corresponding relaxed bed geometry (analogous to Fig. 4b). This will give an idea of how far the grounding line position with a fixed geometry lies from the GIA counterpart. The spin-up could start from a 2 km block of ice (as currently performed) or perhaps from a thin layer of ice that grows until equilibrium (Pattyn et al, 2012). In total, this would only introduce 32 additional simulations, feasible considering this idealised domain.

- **Lithospheric thickness $T_e$.**

  I find striking the steady-state bed displacement sensitivity to lithospheric thickness. To my understanding, this parameter mainly dictates the horizontal extent and the transient response of the vertical displacement, thus only driving minor changes in the steady-state displacement (Coulon et al., 2021; Swierczek-Jereczek et al., 2025). Figure S2 shows the contrary: the equilibrium displacement is apparently dictated by $T_e$. Why is this? I encourage the authors to elaborate on such behaviour, seemingly contradicting previous studies. The consequences of this behaviour are notable, since the reported R-tipping is therefore highly sensitive to $T_e$. I would suggest moving Fig. S2 to the main text since the ultimate physical mechanism is therein illustrated.

- **Shift in the critical melt rate threshold.**

  The study defines the shift relative to the fixed-bedrock geometry $m_{B-\text{tip,fixed}}$ (Eq. 2 and Fig. 6). Nevertheless, one must look at simulations with identical parameter choice to illustrate the contribution of forcing rates (i.e., $T_e$ and $\eta$). Hence, Eq. 2 must be modified to reflect that the threshold shift is compared between simulation with identical GIA parameter values and different melt rates:

$$\Delta m_{R-\text{tip}}(T_e, \eta) = \frac{m_{R-\text{tip}}(T_e, \eta) - m_{B-\text{tip}}(T_e, \eta)}{m_{B-\text{tip}}(T_e, \eta)} \tag{1}$$

  With this new definition, the threshold shift only reflects the contribution of the rate of forcing, rather than the influence of considering bedrock displacement.

- **GIA domain and boundary conditions.**

  The horizontal scale of the bed displacement might reach beyond the edges of the domain. If boundary conditions impose a null vertical displacement at the edges, it would introduce a bias (particularly relevant at large $T_e$).

  Ideally, the vertical displacement should vanish near the edges to avoid this bias. I suggest an easy test to quantify this potential error by simply evaluating the horizontal gradient of the vertical displacement. If these are significantly non-zero as the edge is approached, the GIA domain definition should be extended further, specially along the $y$-axis. If boundary conditions are different (e.g., Robin-type $\partial u/\partial \mathbf{n} + u = 0$), the choice should be justified and detailed in the text.

- **Additional stabilizing feedbacks.**

  The are additional stabilizing feedbacks that could be discussed in the manuscript. For example, heterogeneous precipitation (Fig. 2 in Sergienko, 2022) and the temperature dependency of ice viscosity (Fig. 6 in Moreno-Parada et al., 2025). Sergienko (2022) demonstrated that the feed-back between precipitation, atmospheric surface temperature and ice-sheet surface elevation provides additional stability. Moreno-Parada et al. (2025) showed that the bistability of the system is increased (i.e., a widen hysteresis loop) if the modelled ice temperature and viscosity are fully coupled, due to velocity corrections as the thermal structure adjusts to the changing ice thickness. Both mechanisms are absent in this study, since ice softness and surface accumulation are assumed to be constant.

- **Grid resolution.**

  All simulations are run with $\Delta x = 2$ km following Feldmann and Levermann (2023). Have you performed a convergence analysis where bedrock displacement is included? Given that the model domain is idealised, higher resolutions are feasible. I would suggest taking two parameter choices of the pair $T_e$ and $\eta$ (corresponding to the presence and absence of R-tipping behaviour) to perform a number of simulations for, at least, $\Delta x = 0.5, 1, 4$ km. This would be ideal to compare both the hysteresis loops and the forcing rates exhibiting R-tipping as a function of $\Delta x$, ultimately justifying (or not) the resolution choice if bedrock displacement is considered. The convergence test will further provide robustness to the results.

**2 Minor comments**

- For the self-sustained oscillations, it would be informative to estimate if a parameter combination of $\eta$ and $T_e$ characterizing the Laurentide Ice Sheet gives rise to such an oscillatory behaviour.

- The caption of Figure 8 should read: "*The two dotted horizontal lines mark the range of retrograde bed slope for a fixed bed.*"

- Figure 9: "*reverse MISI*". I would simply use MISI. The instability does not imply a specific direction, but rather to the absence of stable equilibrium positions over retrograde slopes (Schoof, 2007).

**References**

- Swierczek-Jereczek, J., Blasco, J., Robinson, A., Alvarez-Solas, J., and Montoya, M.: Rate-induced tipping in marine-based regions of the Antarctic ice sheet, `https://doi.org/10.21203/rs.3.rs-6701284/v1`, iSSN:2693-5015, 2025.

- Pattyn, F., Schoof, C., Perichon, L., Hindmarsh, R. C. A., Bueler, E., de Fleurian, B., Durand, G., Gagliardini, O., Gladstone, R., Goldberg, D., Gudmundsson, G. H., Huybrechts, P., Lee, V., Nick, F. M., Payne, A. J., Pollard, D., Rybak, O., Saito, F., and Vieli, A.: Results of the Marine Ice Sheet Model Intercomparison Project, MISMIP, The Cryosphere, 6, 573–588, `https://doi.org/10.5194/tc-6-573-2012`, 2012.

- Coulon, V., Bulthuis, K., Whitehouse, P. L., Sun, S., Haubner, K., Zipf, L., and Pattyn, F. (2021). Contrasting response of West and East Antarctic ice sheets to

glacial isostatic adjustment. Journal of Geophysical Research: Earth Surface, 126, e2020JF006003. `https://doi.org/10.1029/2020JF006003`

- Schoof, C. (2007), Ice sheet grounding line dynamics: Steady states, stability, and hysteresis, J. Geophys. Res., 112, F03S28, doi:10.1029/2006JF000664.

- Feldmann, J. and Levermann, A.: Timescales of outlet-glacier flow with negligible basal friction: theory, observations and modeling, The Cryosphere, 17, 327–348, `https://doi.org/10.5194/tc-17-327-2023`, publisher: Copernicus GmbH, 2023.

- Swierczek-Jereczek, J., Montoya, M., Latychev, K., Robinson, A., Alvarez-Solas, J., and Mitrovica, J.: FastIsostasy v1.0 – a regional, accelerated 2D glacial isostatic adjustment (GIA) model accounting for the lateral variability of the solid Earth, Geosci. Model Dev., 17, 5263–5290, `https://doi.org/10.5194/gmd-17-5263-2024`, 2024.

- Sergienko, O.V. No general stability conditions for marine ice-sheet grounding lines in the presence of feedbacks. Nat Commun 13, 2265 (2022).`https://doi.org/10.1038/s41467-022-29892-3`

- Moreno-Parada, D., Robinson, A., Montoya, M., and Alvarez-Solas, J.: Description and validation of the ice-sheet model Nix v1.0, Geosci. Model Dev., 18, 3895–3919, `https://doi.org/10.5194/gmd-18-3895-2025`, 2025.